# Leukemia-intrinsic determinants of CAR-T response revealed by iterative in vivo genome-wide CRISPR screening

Azucena Ramos[1,2,13], Catherine E. Koch[1,2,13], Yunpeng Liu-Lupo[1,2,13], Riley D. Hellinger [1], Taeyoon Kyung[1,3], Keene L. Abbott [1,2], Julia Fröse[1], Daniel Goulet[1], Khloe S. Gordon[1,3], Keith P. Eidell[1], Paul Leclerc[1], Charles A. Whittaker[1], Rebecca C. Larson[4,5], Audrey J. Muscato [6,7], Kathleen B. Yates [6,7], Juan Dubrot [6,7,12], John G. Doench [7], Aviv Regev[2,7,8,9], Matthew G. Vander Heiden [1,2,10], Marcela V. Maus [4,5,6,7,11], Robert T. Manguso [5,6,7], Michael E. Birnbaum [1,3] & Michael T. Hemann [1,2] ✉

CAR-T therapy is a promising, novel treatment modality for B-cell malignancies and yet many patients relapse through a variety of means, including loss of CAR-T cells and antigen escape. To investigate leukemia-intrinsic CAR-T resistance mechanisms, we performed genome-wide CRISPR-Cas9 loss-of-function screens in an immunocompetent murine model of B-cell acute lymphoblastic leukemia (B-ALL) utilizing a modular guide RNA library. We identified IFNγR/JAK/STAT signaling and components of antigen processing and presentation pathway as key mediators of resistance to CAR-T therapy in vivo; intriguingly, loss of this pathway yielded the opposite effect in vitro (sensitized leukemia to CAR-T cells). Transcriptional characterization of this model demonstrated upregulation of these pathways in tumors relapsed after CAR-T treatment, and functional studies showed a surprising role for natural killer (NK) cells in engaging this resistance program. Finally, examination of data from B-ALL patients treated with CAR-T revealed an association between poor outcomes and increased expression of JAK/STAT and MHC-I in leukemia cells. Overall, our data identify an unexpected mechanism of resistance to CAR-T therapy in which tumor cell interaction with the in vivo tumor microenvironment, including NK cells, induces expression of an adaptive, therapy-induced, T-cell resistance program in tumor cells.

Immunotherapies have emerged as crucial components of cancer treatment, rapidly becoming the standard of care for a broad range of malignancies[1]. One of the most promising of these agents today is the adoptive cell transfer (ACT) of autologous T-cells engineered to express chimeric antigen receptors (CARs)[2,3]. Functionally, CARs redirect the cytotoxicity of immune cells towards a patient's tumor. Groundbreaking trials in relapsed B-cell malignancies demonstrated

extraordinary initial efficacy, with upwards of 90% of patients experiencing complete responses (CR)[4–8]. However, despite impressive initial results, long-term follow up of CAR-T treated patients suggests that relapse is a significant problem. Studies have shown that while the median overall survival after CAR-T therapy is on the order of 12 to 20 months, upwards of 60% of patients will experience disease recurrence[5,8–13] Furthermore, up to 20% of patients with B-ALL never

achieve remission and applications of CAR-T cells to other CD19[+] B-cell malignancies, such as chronic lymphocytic leukemia (CLL) and diffuse large B-cell lymphoma (DLBCL), have significantly lower CR rates[5,8,11,14]. Thus, gaining a more complete understanding of the factors governing response to CAR-T therapy will be critical for improving clinical efficacy through superior CAR-T cell design or combination treatment strategies.

To date, a limited number of CAR-T resistance and relapse mechanisms have been described[15]. In some settings, treatment failure has been attributed to early loss of the infused CAR-T cell product, particularly in patients who never achieve remission[14,16]. This has led to the hypothesis that intrinsic CAR-T cell dysfunction is a central determinant of treatment failure due to factors such as the quality of harvested T-cells or variations in production and manufacturing. Alternatively, tumor cell-intrinsic alterations have also been associated with resistance, with target antigen loss arguably garnering the most attention[17,18]. For anti-CD19 CAR-T therapy, various mechanisms of CD19 loss have been reported, including mutations in the CD19 locus, alternative splicing of CD19 mRNA, and lineage switching[14,19–21]. In this setting, as many as 1 in 4 patients relapse with CD19− disease[22,23]. Interestingly, patients can also relapse with CD19+ disease, but it is unclear whether intrinsic CAR-T dysfunction or intrinsic tumor cell alterations are responsible. One recent study in particular provides support for the latter and demonstrated that impaired death receptor signaling in tumor cells led to CAR-T therapy resistance[24]. Gene expression analysis of patient tumor samples prior to CAR-T treatment demonstrated that expression of death receptor pathway genes differed between patients who ultimately achieved CR and those who did not. This pre-treatment tumor expression profile was then able to successfully predict outcomes in an independent patient cohort suggesting that in some cases, CAR-T therapy resistance may arise from pre-existing tumor cell alterations.

Other studies further support the importance of tumor cell-intrinsic alterations driving CAR-T resistance and have highlighted an important role for IFNγ, but the mechanism appears to vary depending on the context. For example, an in vitro genetic screen in a human glioblastoma line showed that loss of interferon-gamma receptor (IFNGR) mediated resistance to CAR-T therapy; this was validated in multiple solid tumor cell lines and xenograft models, and demonstrated that impaired IFNγR signaling didn't have the same effect in lymphoid tumors[25]. Complementary work from the same group demonstrated that CAR-T cells knocked out for IFNγ production had equivalent efficacy in lymphoid xenograft models of leukemia and lymphoma[26]. Other work has described an indirect role of CAR-T produced IFNγ in enhancing anti-tumor immunity. Specifically, IFNγ released from CAR-T cells was shown to remodel the tumor immune landscape and promote a more activated and less suppressive tumor microenvironment[27]. Another recent study, using only CD4 + CAR-T cells, suggested that leukemia cells could be killed indirectly via IFNγ secretion, even when CAR-T cells did not directly engage with the targets[28]. In contrast, in immunocompetent mouse models of pancreatic cancer and melanoma, loss of IFNγR/JAK/STAT signaling sensitized tumor cells to both checkpoint and CAR-T therapy[29,30]. Overall, these conflicting reports highlight the complexity and context-dependence of the relationship between tumor intrinsic alterations and CAR-T resistance, especially when it comes to IFNγ pathway signaling. Resistance mechanisms, particularly those driven by tumor intrinsic alterations, may be particularly influenced by the models tested, whether in vitro, in vivo in immunodeficient models, or in vivo in immunocompetent models, or in patients.

To systematically investigate mechanisms of leukemia-intrinsic resistance to CAR-T therapy in an unbiased manner in an immunocompetent model, we performed parallel, whole-genome in vitro and in vivo CRISPR/Cas9-mediated loss-of-function (LOF) screens in a transplantable mouse model of *BCR-ABL[+]* B-cell acute lymphoblastic leukemia (B-ALL)[31,32]. Here, we utilized an iterative screening approach with a first-of-its-kind modular single guide RNA (sgRNA) library and subsequent validation library. Importantly, this pipeline represents a novel approach to CRISPR/Cas9 screening, particularly for in vivo settings. Analysis of in vivo screening data and gene expression data from CAR-T treated relapsed B-ALL cells identified multiple modulators of IFNγR/JAK/STAT signaling as key mediators of CAR-T resistance. Further characterization of our screening and transcriptional data implicated Qa-1[b] (encoded by *H2-T23*), the murine homolog of Human leukocyte antigen E (HLA-E) and a downstream target of IFNγR/JAK/STAT signaling, in promoting tumor cell resistance to CAR-T therapy. Treatment of leukemic mice with blocking antibodies that prevent interaction of Qa-1[b] with its inhibitory receptor significantly extended survival when combined with CAR-T therapy. Additionally, we observed that depletion of natural killer (NK) cells significantly extended survival of leukemic mice treated with CAR-T therapy, suggesting that NK cell presence can inhibit CAR-T efficacy. These findings suggest a microenvironment-induced "adaptive" resistance mechanism to cell-based immunotherapy and highlight new approaches to enhance CAR-T cell efficacy by altering innate immune signaling and CAR-T/tumor cell interaction without modifying existing CAR-T cell products.

## Results

### A fully immunocompetent mouse model of BCR-ABL1[+] B-ALL enables parallel in vivo and in vitro genome-wide screens for CAR-T resistance

Using an established mouse model of *BCR-ABL1[+]* B-ALL with a high engraftment rate in immunocompetent, syngeneic recipient mice, we engineered Cas9-expressing cells (Cas9[+]) (Supplementary Fig. 1a) with high cutting efficiencies (Supplementary Fig. 1b) for use in unbiased, genome-wide screens[31–35]. To determine if an in vivo screen for CAR-T resistance using immunocompetent mice was tractable, we examined in vivo growth in both wildtype (WT) and Cas9[+] cells, reasoning that if Cas9 expression induced any immunogenic barrier it would manifest as delayed growth kinetics over time. A luciferase[+] Cas9[+] clone (20.12), growth matched to its WT parental line in vitro, was transplanted into non-irradiated immunocompetent male C57BL/6 J (B6) mice. Compared to the WT parental line, no significant growth delays in Cas9[+] cells were detected in vivo in any hematopoietic organ assayed (Supplementary Fig. 1c). Parallel experiments were also performed in non-irradiated male B6 mice and immunocompromised NOD-SCID/IL2Rg[−/−] (NSG) mice. If Cas9 was immunogenic in recipient mice, immunosuppressed NSG mice transplanted with Cas9[+] cells would have succumbed to disease faster than immunocompetent B6 mice transplanted with the same cells. However, no differences in disease latency were observed in repeated experiments (Supplementary Fig. 1d). We note that this lack of immunogenicity may be due the absence of specific host recognition of exogenous Cas9 or to altered antigen presentation pathways in this model, either of which were deemed acceptable in the context of screening for resistance to CAR-T cells.

We then examined the ability of murine CAR-T cells to suppress tumor cell growth in vivo. To mimic lymphodepletion regimens used in patients, we subjected recipient mice to an irradiation-based lymphodepletion protocol prior to tumor transplantation and subsequent CAR-T treatment[36–38]. To calibrate the dose of CAR-T cells, we performed a series of in vivo dosing experiments utilizing CD28-based, 2nd generation murine CARs targeting murine CD19 (mCD19). Previous groups utilizing a similar construct successfully suppressed disease with CAR-T dose ranges in the millions of cells ($5 \times 10^6$ to $2 \times 10^7$) per animal[36,37,39]. In our model, doses of $7 \times 10^6$ to $1 \times 10^7$ CAR-T cells per animal administered two days after the transplantation of $6.0 \times 10^5$ B-ALL cells achieved significant dose-dependent life extension (Fig. 1a). To simultaneously monitor disease suppression in real time, we made

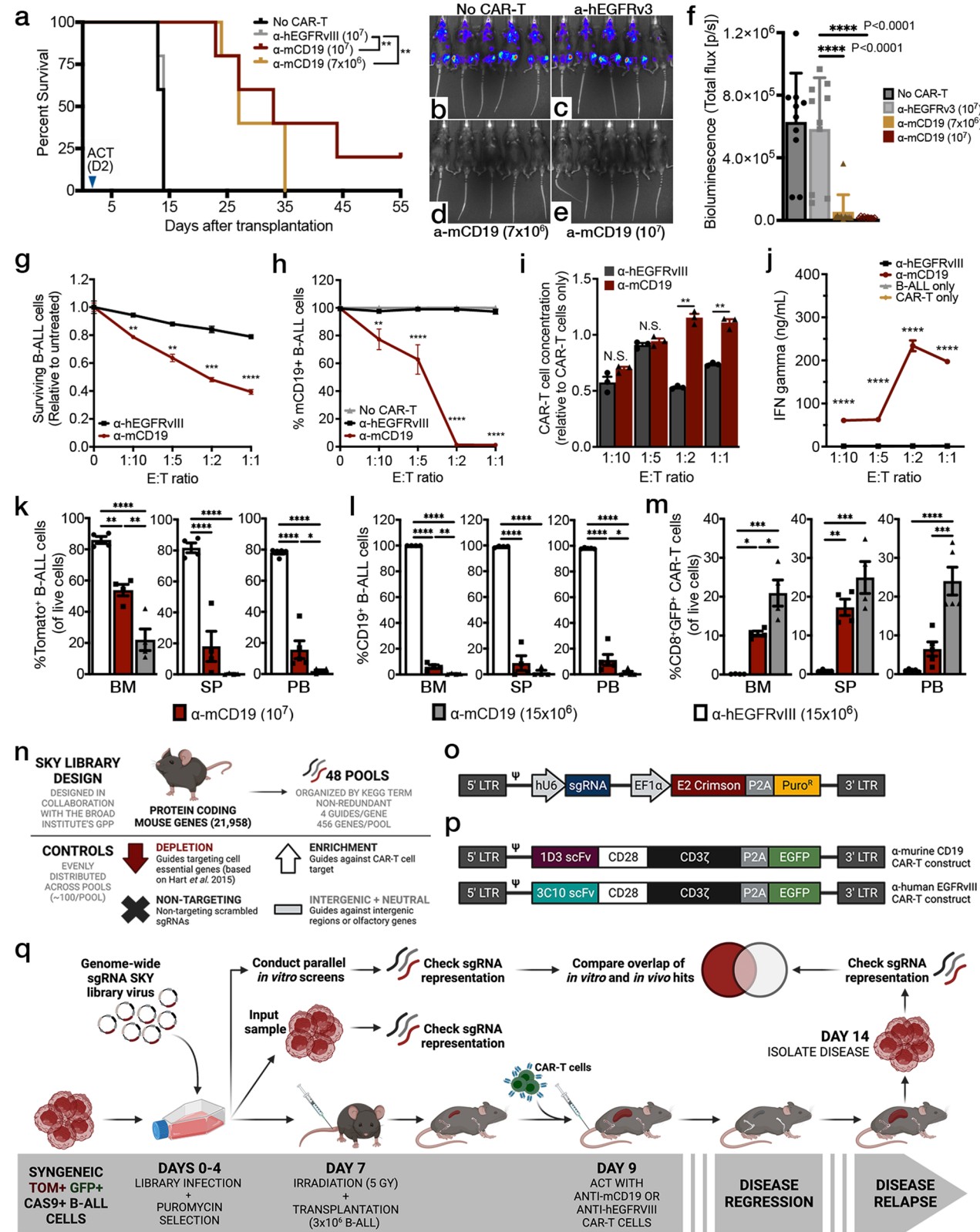

use of firefly luciferase expressed in our leukemia model[34]. Bioluminescence imaging completed at multiple time points after ACT demonstrated that anti-mCD19 CAR-T cells could significantly suppress disease over time (Fig. 1b–f).

To ensure CAR-T functionality, we conducted concurrent matched in vitro cytotoxicity assays for each independent experiment and measured CAR-T expansion and IFNγ (a cytokine released by T cells in proinflammatory conditions) in the resulting culture supernatant. Interestingly, B-ALL cell numbers could be significantly reduced but never fully eliminated in vitro even at very high effector-to-target (E:T) ratios (Fig. 1g). CAR-T cells also induced a rapid and dramatic loss of the mCD19 target epitope on the surface of B-ALL cells (Fig. 1h), a phenotype that our results suggest is target antigen-independent and not a unique feature of our murine B-ALL model (Supplementary

**Fig. 1 | A fully immunocompetent mouse model of BCR-ABL1⁺ B-ALL enables parallel in vivo and in vitro genome-wide screens for CAR-T resistance.**
**a** Survival curves of irradiated B6 mice inoculated with B-ALL cells and treated with indicated CAR-T cell type. Both significant P values displayed = 0.0035. Bioluminescence imaging four days after adoptive cell transfer (ACT) in **b** untreated mice, **c** mice treated with $10^7$ control CAR-T cells, **d** treated with $7 \times 10^6$ anti-mCD19 CAR-T cells, or **e** treated with $10^7$ anti-mCD19 CAR-T cells, and quantified in (**f**), both significant P values displayed <0.001; $n = 10$ mice per group from two independent experiments). **g** In vitro cytotoxicity assays show significant depletion of tumor cells along with **h** target epitope loss when B-ALL cells are treated with anti-mCD19 CAR-T cells at increasing effector to target cell (E:T) ratios. **i** Concomitantly, CAR-T cells expand and **j** release IFNγ when co-cultured with mCD19+ B-ALL cells. For (**g**–**j**), $n = 3$ biologically independent samples per group except for (**j**) where $n = 6$ biologically independent samples for the B-ALL only group. **k**–**m** Experiments to determine the appropriate CAR-T cell dose for either the bone marrow (BM) or spleen (SP) using flow cytometry analysis. Peripheral blood (PB) was also assessed ($n = 4$ mice per group). **l** Target epitope loss and **m** significant CAR-T cell persistence was also observed in all of the organs harvested from mice treated with anti-mCD19 CAR-T cells. **n** Schematic showing the overall design and **o** lentiviral backbone used to create the SKY library. **p** Retroviral vectors encoding the 1D3 single chain variable fragment (scFv) targeting mCD19 (top) and the 3C10 scFv targeting hEGFRvIII (bottom). **q** Diagram of screening layout. Data are mean ± s.e.m. All experiments were repeated at least twice with representative data shown. The significance for survival experiments was determined using log-rank tests. For all other experiments, significance is determined using unpaired two-sided student's t-tests with Bonferroni correction for multiple comparisons, or using one-way ANOVA with Tukey's correction for multiple comparisons when more than two groups were compared. *$P < 0.05$; **$P < 0.01$; ***$P < 0.001$; ****$P < 0.0001$. Exact $P$ values for each comparison shown in (**g**–**j**) and (**k**–**m**) can be found in Supplementary Data 3.

Fig. 1e). Importantly, while antigen loss was a primary phenotype in our B-ALL cells, our cytotoxicity assays still resulted in activated CAR-T cells that expanded (Fig. 1i) and released significant IFNγ after being co-cultured with cells expressing their target antigen (Fig. 1j).

Our group has previously shown that this particular mouse model of B-ALL is highly amenable to in vivo LOF screens[33,34]. Thus, we next sought to determine the appropriate CAR-T cell dose to use for in vivo screening[33,34]. Irradiated B6 mice were transplanted with B-ALL cells (clone 20.12), treated with varying CAR-T cell doses, and monitored using bioluminescence imaging. Animals were sacrificed at peak disease suppression, which occurred on or before day three after CAR-T treatment (Supplementary Fig. 1f). Total disease burden, target tumor antigen expression, and CAR-T expansion were assayed in various hematopoietic organs. We aimed for 80-90% disease suppression, as this significant, but incomplete, level of tumor cell reduction would allow for identification of alterations that could sensitize tumor cells to therapy but would not completely eradicate them. For the bone marrow compartment, this was accomplished using $1.5 \times 10^7$ CAR-T cells, while the splenic and peripheral blood compartments only required a dose of $1 \times 10^7$ CAR-T cells (Fig. 1k). Relapsed disease harvested from CAR-T treated mice showed a striking dose-dependent loss of mCD19 surface expression along with significant CAR-T engraftment in all organs examined (Fig. 1l, m), a consistent phenotype observed across all in vivo experiments. Given that we also observed significant antigen loss and concomitant CAR-T cell persistence in relapsed animals (Supplementary Fig. 1g), we reasoned that this phenotype was due to ongoing CAR-T surveillance. Indeed, when mCD19- leukemia cells were harvested from relapsed mice and cultured in vitro, mCD19 expression was restored as CAR-T cells were depleted from culture (Supplementary Fig. 1h, i). Notably, clinical data has demonstrated that as many as half of all leukemia patients treated with CAR-T relapse with antigen-negative disease. Thus, our system may be modeling this phenomenon[5,9,20].

## Unbiased CRISPR/Cas9-mediated screen identifies genes and pathways involved in in vivo resistance to anti-CD19 CAR-T therapy

Having established a mouse model of anti-mCD19 CAR-T treatment response, we next sought to develop a CRISPR/Cas9 library that would enable genome-wide in vivo screens across a broad range of tumor models with varying engraftment rates. We generated a novel, genome-wide pooled sgRNA library cloned into an optimized lentiviral backbone containing a crimson fluorophore and puromycin selection marker (Fig. 1n–p). The library targets each protein-coding gene (a total of 21,958 genes) in the mouse genome with 4 different sgRNAs per gene. Genes are organized by KEGG term and evenly distributed into 48 individual, non-redundant sub-pools (~456 genes/sub-pool). All sgRNAs targeting a given gene are present in the same sub-pool and control sgRNAs are evenly distributed across all sub-pools. This unique feature allows for sub-pools to be used as stand-alone screening libraries that can also be combined into larger sgRNA pools, depending on the model system and its in vivo engraftment rate. Given the high in vivo engraftment rate of our B-ALL model, we were able to pool our 48 sub-pools into groups of eight, limiting our entire genome-wide in vivo and in vitro screens to six separate screens completed over two experiments (Fig. 1q).

To query factors responsible for CAR-T resistance, we subjected library-infected Cas9⁺ B-ALL cells (20.12) grown in vitro or in vivo to either anti-mCD19 CAR-T therapy or control CAR-T cells targeting human EGFRvIII (hEGFRvIII), which is not expressed in mice or on our B-ALL model. The in vivo arm of our screen followed the layout of our optimized dose finding experiments. We observed that relapsed CAR-T treated mice harbored both CD19-positive and negative disease, as well as persistent CAR-T cell populations in both the spleen and bone marrow (Supplementary Fig. 2a, b, respectively). Concurrent with this in vivo screen, we performed parallel in vitro screens at two effector to target (E:T) cell ratios (Fig. 1q) completed on the same timeline as the in vivo screens. Tumor cells were treated with CAR-T therapy on day 9 and maintained in culture until isolation on day 14. To assay CAR-T functionality during the primary screens, IFNγ release assays were performed after 24 hours of co-culture. In all cases, CAR-T cells released significantly more IFNγ when exposed to their target antigen and no significant differences between experiments could be detected (Supplementary Fig. 2c). On day 14, live, sgRNA-bearing B-ALL cells were isolated from animal organs or cell culture using fluorescence-activated cell sorting (FACS) (Supplementary Fig. 2d). We then used high-throughput sequencing to quantify sgRNA representation in B-ALL cells harvested from each experimental arm. Importantly, we were able to confirm that we achieved extensive coverage of every sgRNA library pool screened (Supplementary Fig. 2e, f). To confirm that our screens were able to effectively interrogate gene function while remaining unaffected by possible artifacts from Cas9-mediated DNA breakage, we quantified the enrichment and depletion of sgRNAs in the control treatment group compared to pre-screen input samples. Indeed, sgRNAs targeting known essential genes significantly depleted in the control treatment groups, whereas those targeting intergenic regions as well as non-targeting sgRNAs did not (Supplementary Fig. 2g).

To assess the genetic dependencies involved in the response of B-ALL cells to anti-mCD19 CAR-T treatment, we compared the anti-mCD19 CAR-T and control treatment arms of the screens across each of the six pools (Fig. 2a). These results were then aggregated to generate a genome-wide perturbation landscape of CAR-T treatment response (Fig. 2b, c). Within such a landscape, sgRNAs that become enriched are targeting genes whose loss of function mediate resistance to CAR-T therapy, whereas those sgRNAs that are most depleted target genes whose loss sensitize leukemic cells to CAR-T therapy. As expected, sgRNAs targeting the *Cd19* locus were the top overall in vivo

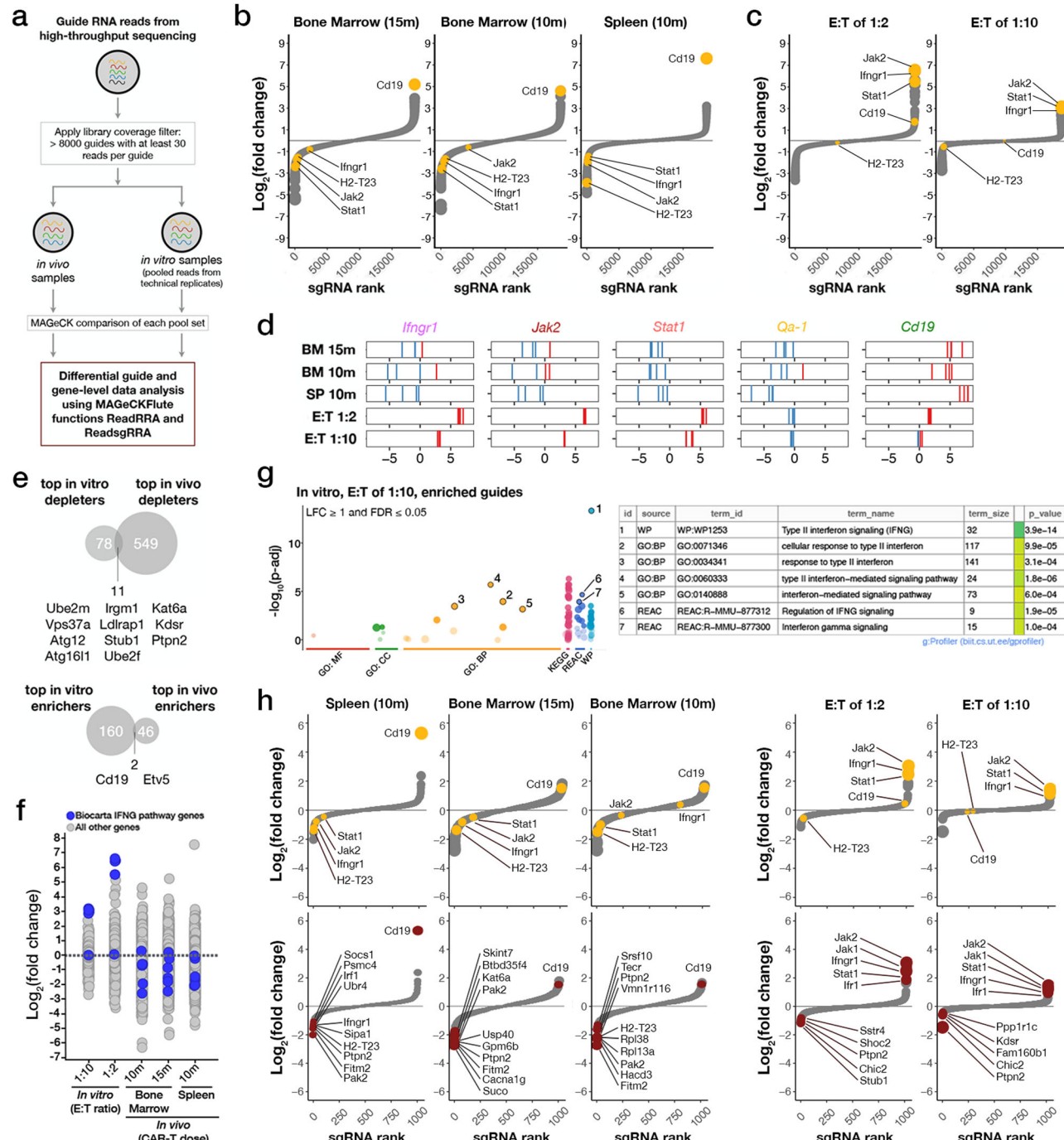

**Fig. 2 | In vivo genome-wide primary and subsequent validation CRISPR-Cas9-mediated knockout screens identify IFNγR/JAK/STAT signaling and antigen presentation pathways involved in resistance to CAR-T therapy. a** Schematic of the data analysis and screen hit discovery workflow. **b**, **c** Waterfall plots of log-fold changes of the representation of sgRNAs against different genes in anti-mCD19 CAR-T cell treated animals **b** or cells **c** compared to anti-hEGFRvIII CAR-T cell treated animals **b** or cells **c** at indicated doses (15 m or $1.5 \times 10^7$ CAR-T cells, 10 m or $1 \times 10^7$ CAR-T cells, effector to target (E:T) ratios of 1:2, or 1:10). For in vivo screens, the organ from which guide-bearing B-ALL cells were collected is also indicated. Genes are ranked by the average log₂fold changes (L2FC) of all sgRNAs against each gene, and point sizes are proportional to the magnitude (absolute value) of L2FC. Waterfall plots display relative biological effect of each hit versus guide ranks. **d** Relative enrichment/depletion of individual guides against top depleting genes *Ifngr1*, *Jak2*, *Stat1*, and *Qa-1^b* are shown in each arm of the screen. Guide RNAs

against *Cd19* are also shown as an indicator of CAR-T treatment pressure. Enriched gRNAs are shown in red. Depleted gRNAs are shown in blue. **e** Venn diagrams showing overlap of top hits from the primary in vitro and in vivo screens. **f** A strip chart representation of the L2FC of sgRNAs targeting genes in the Biocarta_IFNG_Pathway (shown in blue) upon treatment with Cd19 CAR-T cells in vitro (left two columns) or in vivo (right three columns). **g** (Left) A gprofiler plot showing the pathways targeted by the most enriched sgRNAs in the in vitro E:T of 1:10 treatment cohort. (Right) A table showing a list of pathways targeted by the most enriched sgRNAs. **h** Waterfall plots from the validation screen showing the representation of sgRNAs targeting different genes in anti-mCD19 CAR-T cell treated animals or cultured cells compared to control treated organs or cultured cells. Waterfall plots on top show sgRNAs targeting Cd19 and select interferon gamma pathway genes (in yellow). Waterfall plots below show the top ten depleted, or top five depleted/enriched sgRNAs in each context.

enrichers (Fig. 2b–d). Interestingly, a comparison of top hits between the in vivo and in vitro arms of our screen (Fig. 2e) showed little overlap, suggesting that an in vitro co-culture screening modality did not capture key in vivo mechanisms of CAR-T treatment resistance in an immunocompetent model. In fact, pairwise comparisons of hits from different conditions within the same environment, in vivo versus in vivo or in vitro versus in vitro, are highly correlated while comparisons of in vitro to in vivo results show poor correlation (Supplementary Fig. 3a, in vivo vs. in vivo in red, in vitro vs. in vitro in blue, and in vitro vs. in vivo in yellow). Furthermore, certain sgRNAs exhibited opposite behavior in in vivo versus in vitro settings. For example, the top overall enrichers in the in vitro arms of the screen including sgRNAs targeting the IFNγR signaling pathway members Janus kinase 2 (Jak2), signal transducer and activator of transcription proteins 1 (Stat1), and interferon gamma receptor 1 (Ifngr1) instead depleted in the in vivo arms of the screen (Fig. 2b–d). Thus, the loss of IFNγR signaling mediated resistance to CAR-T in vitro but conferred increased sensitivity to CAR-T in vivo. In addition, functional enrichment analysis of screen data using g:Profiler highlighted sgRNAs targeting genes in the Biocarta_IFNG_Pathway as being enriched in vitro but depleted in vivo (Fig. 2f, g and Supplementary Fig. 3b).

Our initial genome-wide pool-based screening approach was essential to represent a whole genome gRNA library in leukemia recipient mice. However, this pooled approach made direct comparisons between sgRNAs across pools difficult. To enable a more definitive ranking of all sgRNAs exhibiting significant biological effects, we selected top depleting sgRNAs from each pool and created a validation library containing six newly generated sgRNAs per gene, targeting a total of 933 genes. We also included 385 non-targeting (scrambled) guides, 100 intergenic cutting control guides, and 135 guides with consistently neutral (neither depleting nor enriching) behavior in our initial screens. In order to minimize any differences due to CAR-T production over the two initial experiments, validation library genes were selected from the top 5% of genes in each pool (divided evenly between enriching and depleting sgRNAs) based on two criteria: 1) sgRNAs exhibiting the largest fold differences in representation relative to input averaged among all animals and 2) the most stably enriched or depleted sgRNAs across all mice in the group, as scored by the coefficient of variation (CV). The final validation list was a union of genes scoring by either criterion over each of the six individual, large pools.

A validation screen completed in a manner identical to the primary screens was then performed. As in our primary screens, guides targeting known essential genes significantly depleted in the control treatment groups relative to all other guides (Supplementary Fig. 3c) and in vivo samples treated with anti-mCD19 CAR-T cells exhibited patterns of sgRNA enrichment and depletion distinct from the control group (Supplementary Fig. 3d). sgRNAs targeting the Cd19 locus were among the top overall enrichers in vivo (Fig. 2h, leftmost three panels), as assessed by FDR values and log2[fold change] (LFC) (Supplementary Data 1). sgRNA targets exhibiting the greatest LFC in all in vivo conditions included Fitm2, Ptpn2, Pak2, And H2-T23 (Fig. 2h, three left lower panels highlighted in red).

Guides targeting the IFNγR/JAK/STAT signaling pathway were once again amongst the top enriching sgRNAs in vitro (with sgRNAs targeting Ifngr1, Irf1, Jak2, and Stat1 all in the top 10 most enriched by FDR and LFC) and the top depleting sgRNAs in vivo (with Ifngr1, Irf1, Jak2, and Stat1 gRNAs in the top 10% most depleted by FDR and LFC), suggesting that this pathway may be a central regulator of response to CAR-T therapy in vivo (Fig. 2h and Supplementary Data 1). Notably, some variation exists between the top LFC depleted sgRNAs in the spleen and bone marrow. Depletion of sgRNAs targeting IFNγR/JAK/STAT signaling components was most dramatic in the spleen, with seven of the top ten LFC depleting guides (targeting Ifngr1, Irf, Socs1, Fitm2, Ptpn2, Pak2,

and H2-T23) all implicated in interferon gamma response. Fitm2 has been identified in other T-cell sensitization screens and has been implicated in interferon gamma signaling, and Ptpn2 and Socs1 are negative feedback regulator of Jak/Stat signaling that has also been shown to promote resistance to T-cell therapy[40,41]. Additionally, Cd19 sgRNA enrichment was more pronounced in spleen versus bone marrow, and Ifngr1 loss yielded no sensitization to CAR-T therapy in the bone marrow of mice injected with 10 million CAR-T cells. These data suggest that there are potential tissue-specific determinants of CAR-T response.

## The IFNγ pathway promotes resistance to CAR-T therapy in vivo and sensitivity in vitro

To further validate our findings, we focused on genes in the IFNγR/JAK/STAT pathway. This pathway was chosen for several reasons. Firstly, components of the IFNγR/JAK/STAT signaling network and downstream targets were significantly depleted in our validation screen. Secondly, while it has been demonstrated that loss of IFNγR/JAK/STAT signaling can promote CAR-T resistance[25], our data suggests that the effect of disrupting this pathway is not necessarily uniform and is instead wholly dependent upon context with respect to the tumor and tissue microenvironment. Thus, our screening system was uniquely poised to dissect relevant and context-dependent IFNγR/JAK/STAT biology. To further validate our findings, we conducted parallel in vivo and in vitro competition experiments (Fig. 3a–c and Supplementary Fig. 4a, b). Here, we used a new, independently generated Cas9-expressing cell line (RH62) from our parental B-ALL model with high cutting efficiency and matched in vitro growth kinetics to our screened clone 20.12 (Supplementary Fig. 4c, d). We focused on genes whose loss sensitized tumors to CAR-T therapy in vivo, as they represent novel drug targets that could potentiate the effects of CAR-T therapy. Using a fluorescence-based competition assay comparing the relative growth of mixtures of isogenic knock out or control B-ALL cells in vitro and in vivo, we found that the proportion of cells null for Ifngr1, Jak2, or Stat1 significantly depleted in both the bone marrow and spleens of mice treated with anti-mCD19 CAR-T therapy (Fig. 3b, c showing log-scale depletion). Notably, no growth differences were observed in mice transplanted with identical cell mixtures treated with control CAR-T cells, indicating that this phenotype was not driven by fitness defects imparted on cells lacking these genes. Additionally, observed results were not related to Cas9 immunogenicity, as Ifngr1−/− B-ALL cells also significantly depleted in transgenic Cas9 mice transplanted with identical isogenic cell mixtures and treated with CAR-T therapy derived from transgenic Cas9 mice (Supplementary Fig. 4e). Interestingly, parallel in vitro experiments again demonstrated contradictory results, with Ifngr1−/−, Jak2−/−, or Stat1−/− cells significantly enriching in the presence of anti-mCD19 CAR-T cells (Supplementary Fig. 4a, b). Ultimately, these data confirm our screen findings that loss of Ifngr1, Jak2, or Stat1 sensitizes B-ALL cells to CAR-T therapy in vivo but promoted resistance in vitro.

To determine whether our findings were specific to leukemia cells, we examined the effect of tumor-intrinsic Ifngr1 deficiency on response to CAR-T cells in a murine model of glioblastoma, GL261. Here, we transduced the extracellular domain of human CD19 (hCD19) into GL261 cells and performed intra-cranial injections into syngeneic recipient mice. As in the B-ALL competition experiment, we mixed fluorescently labeled GL261 cells deficient for Ifngr1 with isogenic WT control cells and tracked the relative behavior of each cell type following injection of anti-hCD19 CAR-T cells. While the magnitude of difference was less than that observed in B-ALL experiments, we observed significant depletion of Ifngr−/− GL261 cells. These data suggest that our results are not limited to leukemia cells (Supplementary Fig. 4f).

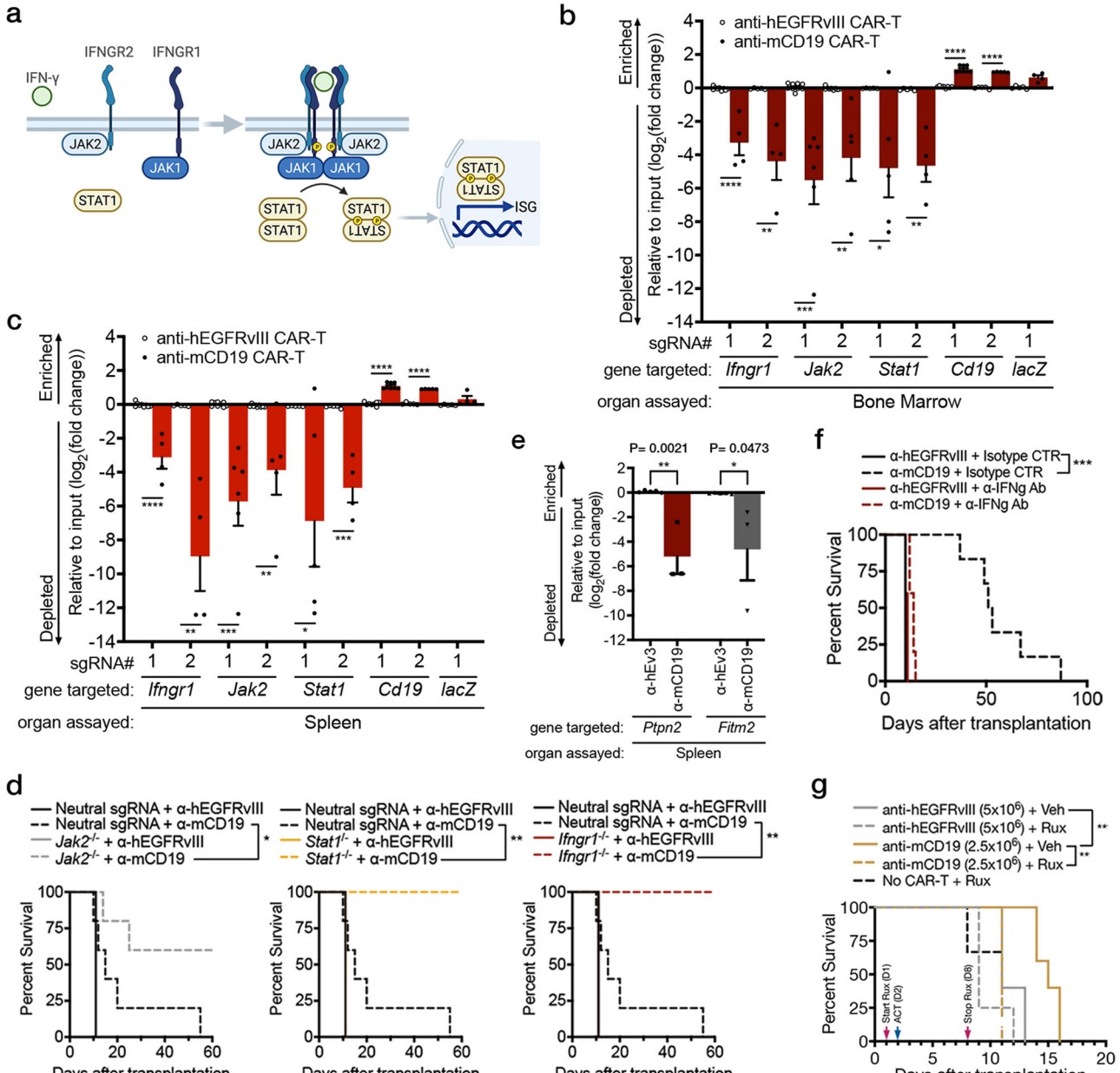

**Fig. 3 | Loss of components of the IFNγR/JAK/STAT pathway sensitizes tumors to CAR-T therapy in vivo. a** Schematic showing the IFNγR signaling pathway. **b**, **c** In vivo competitive assays demonstrate specific depletion of Cas9+ RH62 B-ALL cells lacking components of the IFNγR/JAK/STAT pathway and enrichment of B-ALL cells lacking mCD19 after treatment with anti-mCD19 CAR-T cells in **b** the bone marrow and **c** spleen. For **b**, **c**, n = 9 for the groups *Ifngr1* sgRNA#1, *Jak2* sgRNA#1 and 2, treated with anti-hEGFRvIII CAR-T cells; n = 8 for the groups *Cd19* sgRNA#1 treated with anti-hEGFRvIII or anti-mCD19 CAR-T cells; n = 6 for the group *Jak2* sgRNA#1 treated with anti-mCD19; n = 4 for the groups *Ifngr1* sgRNA#1 and 2, *Stat1* sgRNA#2, and *lacZ* sgRNA#1, treated with anti-mCD19 CAR-T cells; n = 5 for all other groups. **d** Kaplan-Meier curves showing survival in immunocompetent mice transplanted with B-ALL cells deficient in the indicated IFNγR/JAK/STAT pathway member. **e** In vivo competitive assays demonstrate specific depletion of Cas9+ RH62 B-ALL cells lacking Ptpn2 or Fitm2 after treatment with anti-mCD19 CAR-T cells in the spleen. Data shown is from flow cytometry analyses examining the

proportion of live B-ALL cells that are APC+ and therefore, also guide-bearing. In these experiments, n = 5 mice in groups treated with anti-hEGFRvIII CAR-T cells and n = 3 mice in groups treated with anti-mCD19 CAR-T cells. **f** A Kaplan–Meier curve showing overall survival in leukemia-bearing mice treated with 2.5 × 10⁶ control or anti-mCD19 CAR-T cells in the presence or absence of blocking IFNγ antibody. **g** A Kaplan-Meier curve of overall survival in leukemia-bearing mice treated with control or 2.5 × 10⁶ anti-mCD19 CAR-T cells in the presence or absence of Ruxolitinib, a JAK1/2 inhibitor. In vivo competition assays were repeated three times. Survival experiments were completed twice. Pharmacologic studies using anti-IFNγ blocking antibodies or JAK inhibitor were completed twice. The significance of survival experiments was determined using log-rank tests. For all other experiments, significance was determined using unpaired two-sided student's t-tests with Bonferroni correction for multiple comparisons. Data are mean ± s.e.m. *P < 0.05; **P < 0.01; ***P < 0.001; ****P < 0.0001. Exact P values for each comparison shown in (b-d) and (f-g) can be found in Supplementary Data 3.

Next, to further validate our in vivo B-ALL results, we generated pure, clonal populations of leukemic cells deficient in *Ifngr1*, *Jak2*, or *Stat1* (Supplementary Fig. 5a). Consistent with our previous data, mice transplanted with these cells demonstrated increased sensitivity to anti-mCD19 CAR-T therapy, resulting in significantly

increased survival and complete lack of tumor relapse or detectable disease during the indicated time period following treatment (Fig. 3d and Supplementary Fig. 5b). Thus, these results suggest that tumor cells engage IFNγR/JAK/STAT signaling in vivo to resist CAR-T cell killing. Conversely, in vitro CAR-T efficacy appears to be

at least partially dependent on the ability of tumor cells to sense IFNγ.

## Tumor cells lacking *Ptpn2* and *Fitm2* are highly sensitized to IFNγ-mediated cell death

An interesting exception to the in vitro and in vivo opposing phenotypes witnessed among regulators in response to CAR-T therapy was *Ptpn2*. PTPN2 is a negative feedback regulator of IFNγR/JAK/STAT signaling; loss of PTPN2 leads to sustained IFNγR/JAK/STAT signaling, and drugs to inhibit PTPN2 and amplify tumor responses to checkpoint blockade are in clinical development[42]. In the context of our data, with regard to IFNγR/JAK/STAT loss leading to enhanced susceptibility to CAR-T cells in vivo, sgRNAs targeting *Ptpn2* should have depleted in vitro and enriched in vivo. However, *Ptpn2*-targeting sgRNAs were among the most potently depleted, both in vitro and in vivo, suggesting that loss of PTPN2 conferred enhanced susceptibility to CAR-T cell killing both in vitro and in vivo. To validate these findings, we performed fluorescence-based competition assays comparing the relative growth of mixtures of isogenic *Ptpn2* KO or control B-ALL cells treated with CAR-T therapy in vivo or in vitro and found that *Ptpn2* KO cells depleted in mice treated with anti-mCD19 CAR-T therapy as well as in vitro (Fig. 3e). Given that PTPN2 is a negative feedback regulator of IFNγR/JAK/STAT signaling, we hypothesized that depletion of anti-*Ptpn2* sgRNA expressing cells in vitro may result from hypersensitization of tumor cells to IFNγR signaling. Indeed, we found that *Ptpn2* KO cells were strongly sensitized to IFNγ, even in the absence of CAR-T therapy (Supplementary Fig. 5c). Given that *Ptpn2* sgRNA depletion is only seen in the in vitro screen condition in which anti-mCD19 CAR-T cells were added, these data suggest that mCD19-mediated release of IFNγ from anti-mCD19 CAR-T cells can exert anti-tumor activity in our in vitro screen conditions. Consistent with this hypothesis, sgRNAs targeting additional putative negative regulators of IFNγR signaling were also depleted in the in vitro screen setting. For example, the E3 ubiquitin ligase *Stub1*, a negative regulator of IFNγR signaling, was one of the most depleted sgRNAs in vitro (Fig. 2g). Additionally, sgRNAs targeting *Chic2*, a recently identified binding partner of *Stub1*, that, along with *Stub1*, regulates cytokine signaling, were also among the most depleted sgRNAs in vitro[43]. These data suggest that the Stub1/Chic2 complex may serve as a negative regulator of IFNγR signaling. Overall, these data suggest that, in our in vitro screen setting, tumor cell death predominantly results from high dose IFNγ released from CAR-T cells following mCD19/anti-mCD19 CAR-T interaction.

Like *Ptpn2*, *Fitm2* loss has previously been shown to sensitize tumor cells to T cell-mediated killing[40]. In the case of *Fitm2*, this sensitization has also been attributed to IFNγ-mediated cell death. To determine whether *Fitm2* loss similarly sensitizes our leukemia cells to IFNγ, we generated *Fitm2* KO cells and treated them with IFNγ in vivo and in vitro. *Fitm2* KO cells depleted in mice treated with anti-mCD19 CAR-T therapy (Fig. 3e), and FITM2 loss sensitized cells as robustly as PTPN2 loss to IFNγ-induced cell death (Supplementary Fig. 5c). Thus, at least two of our most depleted screen gRNA targets function by protecting cells from IFNγ-mediated cell death. Notably, a recent study has shown that CAR-T-induced cell death can occur indirectly via IFNγ release from activated CAR-T cells. While our data shows that loss of IFNγR/JAK/STAT signaling generally sensitizes tumor cells to CAR-T therapy in vivo, it also shows that perturbations that lead to amplified or altered IFNγR signaling can, as suggested in previous studies, sensitize tumors to a non-specific death induced by activated T cells[28].

## Global targeting of JAK/STAT signaling does not enhance CAR-T therapy

Having established the IFNγR/JAK/STAT pathway as a key mediator of response to CAR-T therapy, we next explored whether global blocking of IFNγ in vivo could potentiate the effects of CAR-T therapy. To this end, we administered blocking antibodies targeting IFNγ the day

before ACT, and every three days thereafter (Supplementary Fig. 5d). Rather than enhancing antitumor effects, blocking IFNγ in the context of anti-mCD19 CAR-T treatment abrogated the anti-tumor effects of CAR-T cells. Mice treated with anti-mCD19 CAR-T and anti-IFNγ antibody succumbed to disease at the same time as mice treated with control CAR-T cells and an isotype control antibody (Fig. 3f and Supplementary Fig. 5e). Identical results were obtained when animals were co-treated with ruxolitinib (a JAK1/2 inhibitor) and CAR-T therapy in a similar experiment (Fig. 3g and Supplementary Fig. 5f–m).These data are in line with recent reports that showed that knocking out *Ifng* in CAR-T cells significantly impaired their ability to eliminate murine lymphoma cells in vivo[39,44] and from drug screens where co-culture of B-ALL cell lines with CAR-T cells in the presence of JAK/STAT inhibitors reduced the efficacy of CAR-T cells in vitro[39,44]. These data suggest that in immune competent murine models of B-ALL and lymphoma, IFNγ is important for tumor susceptibility to CAR-T cells in vivo and show that global blocking IFNγ did not enhance the in vivo efficacy of CAR-T cells.

## *H2-T23* is an in vivo-specific mediator of resistance to CAR-T therapy

To further explore how IFNγR/JAK/STAT signaling in resistance to CAR-T therapy, we examined screen hits downstream of this pathway with known immunoinhibitory functions. One of the most promising candidate genes was the non-classical class I major histocompatibility complex (MHC-I) gene *H2-T23* which encodes Qa-1b, the murine homolog of human leukocyte antigen E (HLA-E)[45,46]. Surface expression of HLA-E has been shown to be induced by IFNγ signaling and sgRNAs targeting *H2-T23* showed potent depletion in vivo in our primary genome-wide screen[47]. In our validation screen, these guides were once again amongst the top depleting sgRNAs (Fig. 2b, h), suggesting that loss of Qa-1b sensitized tumor cells to CAR-T cells in vivo. In addition, Qa-1b is the only known ligand of the CD94/NKG2A receptor which is expressed on the surface of NK cells and CD8+ T-cells[48–52]. The binding of NKG2A/CD94 to its ligand transmits a signal that inhibits the effector functions of NK and CD8+ T cells (Fig. 4a)[50–54]. Several groups have observed a significant enhancement in the antitumor effects of immune checkpoint blockade (ICB) and cancer vaccines when combined with HLA-E/NKG2A axis blockade. Early clinical trials have also demonstrated encouraging results in a variety of cancers including hematologic malignancies[46,53–55]. Furthermore, Qa-1b was identified as a key mediator of resistance to ICB therapy in an in vivo genome-wide loss of function screen performed using an immunocompetent mouse model of pancreatic cancer[29]. This study also showed that loss of Qa-1b in pancreatic ductal adenocarcinoma (PDAC) cells promotes sensitivity to CAR-T treatment in vitro. Consistent with this data, in a competition assay mixing *H2-T23*+/+ and *H2-T23*−/− PDAC cells that express CD19, we see that loss of Qa-1b sensitizes cells to anti-CD19 CAR-T therapy relative to WT cells (Fig. 4b). Notably, however, the increased sensitivity of *H2-T23*−/− to CAR-T therapy required pretreatment of all cells with IFNγ. Thus, IFNγ induction of Qa-1b on tumor cells prior to CAR-T exposure may be necessary to witness the effect of Qa-1b inhibition on CAR-T anti-tumor activity.

To further examine the consequence of Qa-1b loss on tumor cell response to CAR-T therapy, we generated pure populations of *H2-T23*−/− B-ALL lines by sorting via FACS for bulk sgRNA-bearing cells that could no longer be induced to express Qa-1b (Supplementary Fig. 6a, b)[54]. Consistent with our screen results, *H2-T23*−/− tumors were sensitized to CAR-T therapy in vivo, leading to significant life extension in mice, including an absence of tumor relapse in fully immunocompetent animals (Fig. 4c). Notably, immunodeficient NSG recipient mice bearing *H2-T23*-deficient tumors showed extended survival in response to CAR-T therapy but, unlike B6 recipients, eventually relapsed, suggesting that the full effect of *H2-T23*-deficiency on CAR-T efficacy may require additional components of the

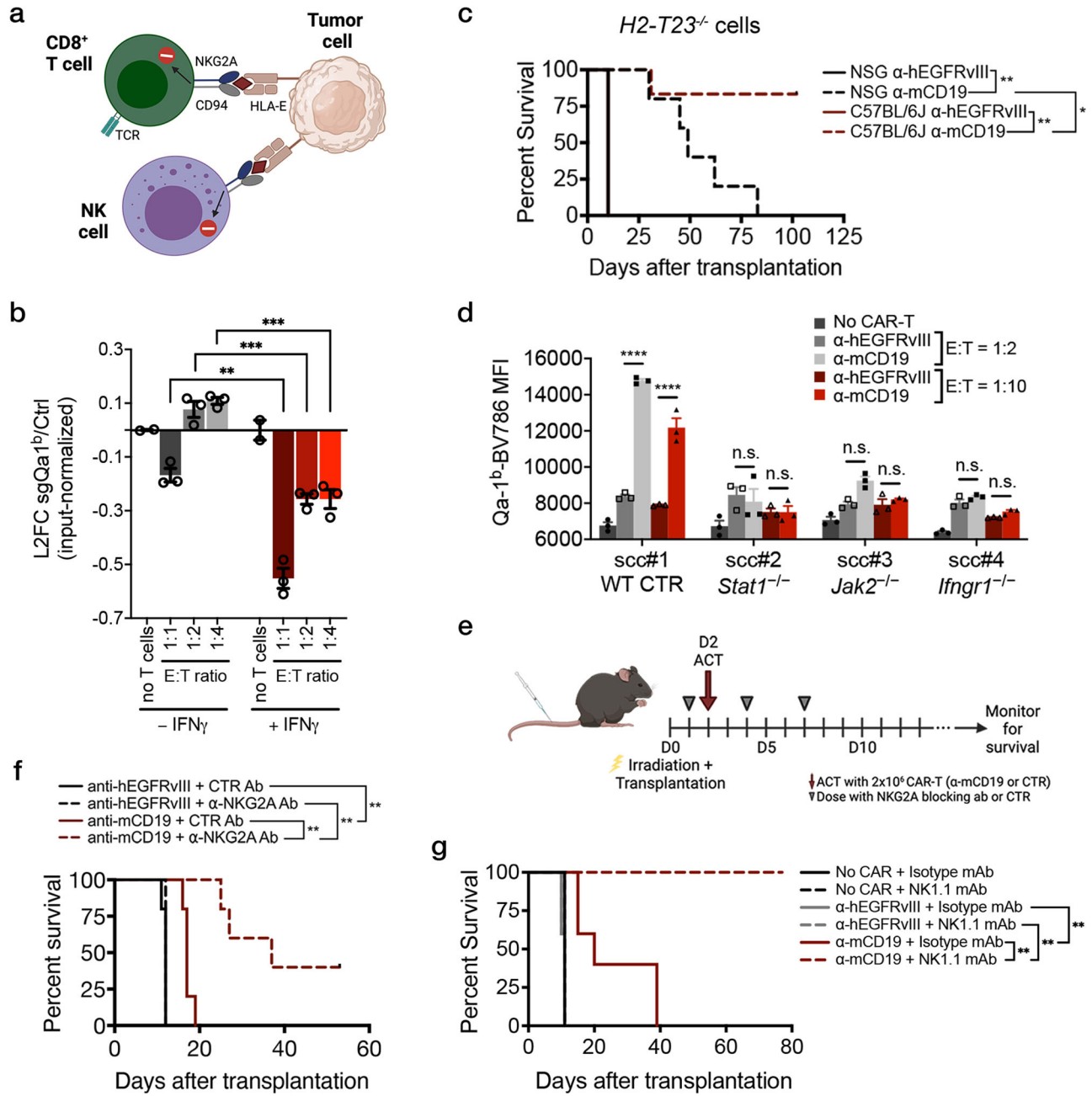

**Fig. 4 | Loss of Qa-1ᵇ, a component of the MHC-I pathway, or pharmacologic blockade of NKG2A, the only known receptor of Qa-1ᵇ, sensitizes B-ALL cells to CAR-T therapy in vivo. a** Schematic showing known effects of the HLA-E/NKG2A/CD94 axis in human NK and CD8⁺ T-cells. **b** In vitro competitive assays where CD19-expressing *H2-T23⁻/⁻* and CD19-expressing *H2-T2⁺/⁺* pancreas cancer cells were mixed in a 50:50 ratio and exposed to the indicated concentration of anti-mCD19 CAR-T cells. The log₂fold change (L2FC) of *H2-T23⁻/⁻* to control cells is shown. Cells were either pre-treated (right) with exogenous recombinant IFNγ or left untreated throughout the experiment (left). For the no T cells groups, n = 2 biologically independent samples; n = 5 for all other groups. **c** Immunocompetent (C57BL/6J) mice transplanted with *H2-T23⁻/⁻* cells show increased survival compared to immunocompromised (NSG) mice transplanted with the same *H2-T23⁻/⁻* cells and treated with anti-mCD19 CAR-T cells (n = 5 mice per group). **d** Mean fluorescence intensity (MFI) of Qa-1ᵇ-Brilliant Violet 786 (BV786) after in vitro CAR-T treatment of single cell clones (scc) deficient in IFNγR/JAK/STAT pathway members at two different E:T ratios. Wildtype scc were also assayed, as shown. **e** Treatment

schedule for the anti-NKG2A blocking antibody experiment. **f** A Kaplan-Meier curve showing overall survival in mice treated with a murine version of the anti-NKG2A antibody, monalizumab or a control antibody, concurrently with mCD19 or control hEGFRvIII CAR-T cells, as indicated. **g** Kaplan-Meier curves showing overall survival in mice treated concurrently with either anti-NK1.1 or isotype control antibodies along with either anti-mCD19, anti-hEGFRvIII, or no CAR-T cells. In this experiment, mice were pre-treated with anti-NK1.1 antibody one day prior to CAR-T therapy. The survival experiments and in vitro CAR-T treatment of *H2-T23⁻/⁻* cells were completed at least twice and were paired each time. Pharmacologic experiments were completed once. The significance of survival experiments was determined using log-rank tests. For all other experiments, significance was determined using unpaired two-sided student's t-tests with Bonferroni correction for multiple comparisons. Data are mean ± s.e.m. \*P < 0.05; \*\*P < 0.01; \*\*\*P < 0.001; \*\*\*\*P < 0.0001. Exact P values for each comparison shown in (**b**–**d**) and (**f**–**g**) can be found in Supplementary Data 3.

adaptive immune system. When tested in vitro using fluorescence-based competition assays as before, no changes in the proportion of *H2-T23*$^{-/-}$ cells could be detected in any treatment group, as demonstrated in our primary screen (Supplementary Fig. 6c). Finally, to investigate whether cells deficient in components of the IFNγR/JAK/STAT pathway were also functionally null for *H2-T23*, we performed in vitro cytotoxicity experiments. While control cells were fully capable of inducing dose-dependent expression of Qa-1$^b$ upon exposure to anti-mCD19 CAR-T cells, clonal cell lines deficient in *Ifngr1*, *Jak2*, or *Stat1* were completely blunted in their ability to express this molecule at all E:T ratios tested (Fig. 4d) or in repeated experiments using recombinant IFNγ in lieu of CAR-T therapy (Supplementary Fig. 6d).

Together these data suggest that IFNγR signaling mediates the induction of Qa-1$^b$ on tumor cells, leading to the engagement of the NKG2A/CD94 receptor on CAR-T cells and subsequent inhibition of CAR-T efficacy. Consistent with this hypothesis, IFNγ pretreatment is necessary to reveal the impact of Qa-1$^b$ loss in co-culture experiments using 1:1 mixtures of control *and H2-T23*$^{-/-}$ PDAC cells as mentioned above (Fig. 4b). These data led us to investigate whether pretreatment of cultured B-ALL cells with IFNγ could similarly protect these cells from CAR-T therapy. We performed in vitro competition experiments using mixed populations of *Ifngr1*$^{-/-}$ and *Ifngr1*$^{+/+}$ cells in the presence and absence of IFNγ pretreatment prior to CAR-T cell exposure (Supplementary Fig. 6e). In the absence of IFNγ pretreatment, *Ifngr1*$^{-/-}$ cells were more resistant than *Ifngr1*$^{+/+}$ cells to CAR-T therapy, consistent with our in vitro screening results. However, pre-exposure of these mixed cultures to IFNγ eliminated the selective resistance of *Ifngr1*$^{-/-}$ cells to CAR-T treatment, likely due to Qa-1$^b$ induction in the *Ifngr1*$^{+/+}$ cells. These data may explain, in part, a major difference in in vitro and in vivo screen conditions, as tumor cells in vivo (but not in vitro) are exposed to IFNγ signaling in the tumor microenvironment, with concomitant induction of Qa-1$^b$, prior to CAR-T administration. Ultimately, this "pre-CAR-T" IFNγ exposure and subsequent Qa-1$^b$ induction may protect tumor cells from CAR-T therapy in vivo, a phenomenon that cannot be captured in vitro unless the in vivo environment is mimicked by pre-treating tumor cells with IFNγ.

To determine whether interruption of Qa-1$^b$/NKG2A inhibitory signaling might have therapeutic potential in the context of CAR-T therapy, we introduced an antibody targeting NKG2A into leukemia-bearing mice concurrently with the introduction of CAR-T cells (Fig. 4e, f and Supplementary Fig. 6f). This antibody blocks Qa-1$^b$-mediated inhibitory signaling through NKG2A. Recipients of the anti-NKG2A antibody showed significantly longer leukemia-free survival relative to control treated animals, including a number of mice failing to relapse over the course of the experiment. Together, these data identify the NKG2A/HLA-E axis as a novel dependency, at least in CAR-T-treated B-ALL cells, and nominate this pathway as a potential target which, when inhibited, may potentiate the efficacy of CAR-T therapy.

### An anti-NK1.1 antibody promotes CAR-T efficacy in vivo

While NKG2A is present on subsets of CD8$^+$ T-cells, it is best characterized as an inhibitory NK cell receptor. Given that *H2-T23* deficient tumor cells relapsed after CAR-T treatment in immunodeficient NSG mice (which lack NK cells) but not in immunocompetent B6 mice possessing NK cells, we hypothesized that NK cells may contribute to anti-leukemia activity in the context of CAR-T therapy. To examine the role of NK cells in this setting, we depleted NK cells in B6 mice using an antibody targeting NK1.1. As expected, treatment of mice with NK1.1 antibody significantly reduced overall NK numbers, as measured by the percentage of NK1.1+ cells 24 hours after antibody injection (Supplementary Fig. 6g). Leukemia-bearing mice were treated with anti-mCD19 or control CAR-T therapy and dosed concurrently with NK1.1 or isotype control antibody. Mice were dosed again with NK1.1 or isotype control antibodies seven days later in order to maintain NK cell

depletion. No difference in survival was observed between mice receiving control CAR-T treatment and NK1.1 antibody or isotype control antibody suggesting that NK depletion alone has no effect on B-ALL disease progression (Fig. 4g and Supp Fig. 6h). Surprisingly, mice receiving anti-mCD19 CAR-T treatment and NK1.1 antibody had improved survival extension compared to mice treated with anti-mCD19 CAR-T therapy and isotype control (Fig. 4g and Supplementary Fig. 6h). This improvement in survival occurred using distinct treatment protocols that differed in the timing of NK1.1 administration (Supplementary Fig. 6i), either injected at the time of CAR-T cell administration or 24 hours prior to CAR-T cell injection. One possible explanation for this is observation is that NK cells are the source of IFNγ that induces Qa-1$^b$ on B-ALL cells in the in vivo leukemia microenvironment. Indeed, NK cells are known to be potent producers of IFNγ at sites of inflammation[56]. While additional experiments must be performed to obtain a definitive answer, these findings reveal a potential therapeutic strategy for improving CAR-T efficacy through combination with NK1.1 antibody. Additionally, these data suggest that Qa-1$^b$ on tumor cells most likely diminishes CAR-T efficacy directly via interaction with NKG2A/CD94 present on CAR-T cells rather than on NK cells.

### An IFNγR/JAK/STAT program delineates cells refractory to anti-CD19 CAR-T therapy

To further explore the transcriptional programs driving B-ALL CAR-T resistance mechanisms we identified, we transcriptionally profiled B-ALL cells after in vivo treatment with CAR-T cells. Principal component analysis of transcription profiles using the most variable genes across samples showed a clear separation of animals treated with anti-mCD19 CAR-T cells from those treated with control anti-hEGFRvIII CAR-T cells (Fig. 5a). Differential expression analysis showed that several genes involved in JAK/STAT signaling and antigen processing and presentation pathways, including *Stat1*, *Irf1*, *Socs1*, and several MHC genes, were highly expressed in samples treated with anti-mCD19 CAR-T therapy compared to those treated with control CAR-T therapy (Fig. 5b, c). Interestingly, sgRNAs targeting these genes were also among the top depleting guide RNAs in our in vivo screens, suggesting that these genes are part of an expression program that may contribute to therapeutic resistance.

Similar to many other tumor types, B-ALL often displays a heterogeneous response to therapy in vivo. To dissect the in vivo transcriptional responses to anti-mCD19 CAR-T therapy at a finer resolution and pinpoint specific cell populations giving rise to relapse, we performed bulk-RNA seq and droplet-based single-cell RNA-seq (scRNA-seq) on a total of 124,523 leukemia cells harvested from the bone marrow and spleen of mice treated with anti-mCD19 or control anti-hEGFRvIII CAR-T cells. Unsupervised clustering revealed 15 distinct cell populations as seen in the 2-dimensional uniform manifold approximation and projection (UMAP) plots (Fig. 5d, left panel). Cells from different treatment arms and tissues occupy distinct clusters (Fig. 5d, right panel and Supplementary Fig. 7a). Interestingly, we did not observe loss of *Cd19* transcript expression in any sample (Supplementary Fig. 7b, left panel), indicating that loss of CD19 surface expression in this context is likely the result of post-translational regulation. Remarkably, a few clusters (2, 4, 5, and 7) were substantially enriched for cells from anti-mCD19 CAR-T treatment groups while being depleted in leukemia cells from the control groups (Fig. 5e). Consistent with our bulk RNAseq data, these clusters were marked by elevated expression of genes in the IFNγR signaling and allograft rejection pathways, as well as MYC and E2F target sets (Fig. 5f). Importantly, several signature genes in cluster 2 were also among the top depleting sgRNAs in our in vivo screens. Figure 5g highlights the magnitude and pervasiveness of the expression of these genes across cell clusters. Additionally, clusters 4, 5, and 7 were enriched for genes involved in OXPHOS and mTORC1 signaling pathways (Fig. 5f). Cells in

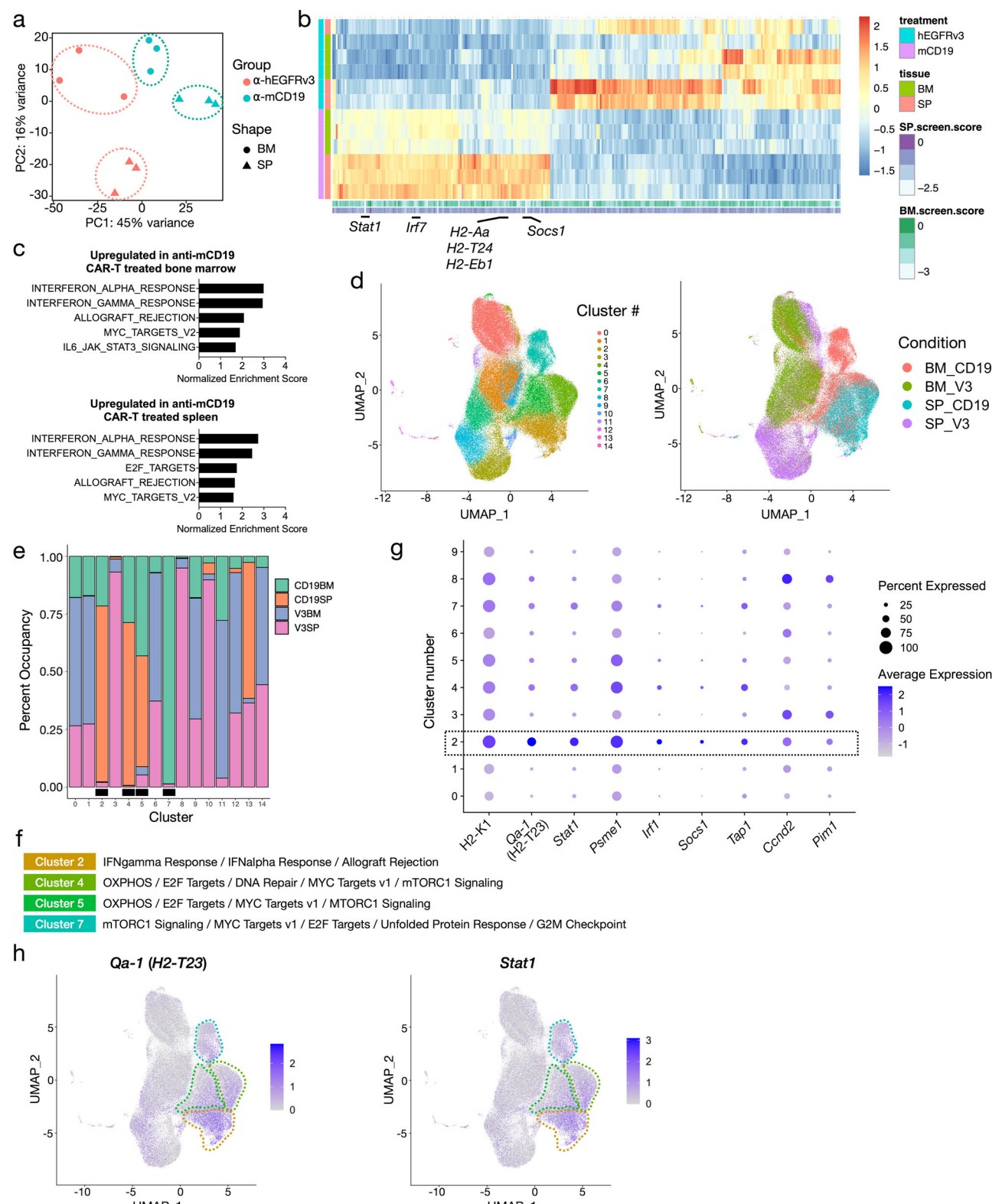

clusters 2 and 4, which contain mostly splenic disease from anti-mCD19 CAR-T treated animals, showed widespread elevated expression of *Stat1* and *H2-T23* (Fig. 5g, h). Intriguingly, transporter associated with antigen processing 1 (*Tap1*) is also over-expressed in these clusters (Supplementary Fig. 7b, right panel). Cluster 7, which is almost entirely comprised of cells from anti-mCD19 CAR-T treated bone marrow, is distinguished by a strong G2/M arrest phenotype (Fig. 5f and Supplementary Fig. 7c). This cluster was also characterized by

elevated expression of *H2-T23* and *Stat1* (Fig. 5h). Transcription factor (TF) motif analysis of gene regulatory regions specifically upregulated in clusters 2, 4, 5, and 7 revealed enrichment of motifs critical for TFs involved in interferon response, such as ETS family TFs, IRF8, and ELK4, with particularly strong enrichment in clusters 2 and 4 (Supplementary Fig. 7d).

Our single-cell and bulk expression data, taken together with results from our in vivo screens and validation assays, further

**Fig. 5 | Bulk and single cell gene expression profiling pinpoints specific cell subsets and expression programs as a source of CAR-T cell resistance in B-ALL. a** PCA plot of bulk RNA-seq profiles of bone marrow (BM) and spleen (SP) samples collected from mice treated with either anti-mCD19 or anti-hEGFRvIII (control) CAR-T cells. **b** Heatmap of genes differentially expressed between anti-mCD19 and anti-hEGFRvIII CAR-T cell-treated samples. Scores of genes from the screen are shown next to each gene's expression profile (column). Top depleting sgRNAs in the BM and SP are labeled. **c** Top Hallmark gene sets enriched among genes overexpressed in mCD19 CAR-T cell therapy compared to control treated BM or SP samples. Bar graphs show normalized enrichment scores for each of the top 5 enriched pathways. Enrichment analysis was performed using GSEA and all pathways shown have FDR < 0.05. **d** 2-dimensional UMAP plots of single cell gene expression profiles collected from mice treated with either anti-mCD19 or anti-hEGFRvIII CAR-T cells. Data are collected from n = 3 mice per treatment group. Left: cells color-coded by clusters discovered through unsupervised clustering.

Right: cells color-coded by treatment and tissue groups. **e** Participation of cells from each tissue/treatment group in each cluster. Bar graphs show percentage of cells in each cluster belonging to each tissue/treatment group (anti-mCD19 CAR-T cell treated BM or SP labeled CD19BM and CD19SP, respectively; anti-hEGFRvIII control CAR-T treated BM or SP labeled V3BM and V3SP, respectively). Clusters where there is substantial enrichment of cells under anti-mCD19 CAR-T therapy are marked with a black box underneath the bar plots. **f** Hallmark gene sets enriched in the 4 treatment-specific clusters as labeled in (**e**). **g** Dot plot showing relative expression patterns of cluster 2 genes which are also among top depleting sgRNAs (L2FC < −1.75) in the BM samples of the in vivo screen. Dot size is proportional to the percentage of cells in each cluster expressing each gene and dot color indicates average expression of each gene in each cluster. **h** UMAP plots showing expression of $Qa\text{-}1^b$ and $Stat1$ in single cell samples, with cells color-coded by expression levels.

support the hypothesis that downstream pathways driven by interferon-gamma receptor signaling, including antigen processing and presentation and the expression of $H2\text{-}T23$, plays a crucial role in dictating B-ALL response to anti-mCD19 CAR-T therapy. Here, a defining characteristic of the resistant tumor microenvironment is leukemia engagement by ongoing CAR-T induced inflammatory signaling.

## High JAK/STAT signaling correlates with CAR-T resistance in humans

Finally, to examine whether JAK/STAT and inflammatory signaling in tumor cells might also promote resistance to CAR-T therapy in human B-cell malignancies, we examined RNAseq data sets generated using pre-treatment bone marrow biopsy samples from patients with B-ALL who received CAR-T therapy[24]. We generated a "sensitizer signature" (gene list found in Supplementary Data 2a) from the overlap of our bone marrow depleting hits and reasoned that decreased expression of this signature should correlate with better outcomes in patients since these genes represent novel resistance mediators to CAR-T therapy. In line with our prediction, patients who experienced complete responses (CRs) show significantly less expression of this "sensitizer signature" compared to non-responders (NRs) (Fig. 6a). To further examine these patient data, we also generated a JAK/STAT/MHC-I resistance signature by overlapping the top depleting sgRNAs in our screen with the top overexpressed genes in tumor cell cluster 2 from our transcriptional analysis. Strikingly, this resistance signature was correlated with poor outcomes, with NRs demonstrating significantly increased expression of our JAK/STAT/MHC-I gene set compared to CRs (Fig. 6b and Supplementary Data 2b). Notably, no correlation was seen between pre-treatment HLA-E gene expression and clinical outcome (Supplementary Fig. 8), perhaps due to the fact that IFNγR/JAK/STAT signaling induces HLA-E peptide loading and cell surface occupancy rather than simply alterations in gene expression. These expression data are consistent with recent reports in large B-cell lymphoma (LBCL) where high intratumoral interferon signaling is correlated with poor outcomes after CAR-T treatment in patients[57]. Here, tumor cell expression of an interferon-stimulated gene resistance signature (ISG.RS) associated with ICB resistance, is a strong predictor for CAR-T treatment failure in LBCL[30,58]. Importantly, this signature was not a general marker of more aggressive leukemias, as it was previously shown to be not associated with resistance to conventional combination chemotherapy regimens[57]. Further examination of this signature in our model revealed that relapsed B-ALL cells with high IFNγR signaling after CAR-T treatment failure were also enriched for expression of the ISG.RS gene set (Fig. 6c). Taken together, these data suggest that intratumoral IFNγR/JAK/STAT signaling, along with downstream antigen processing and presentation pathways, may be key determinants of CAR-T response in human B-ALL (Fig. 6d). Notably, our screens can also nominate novel combination

strategies to re-sensitize tumor cells to CAR-T therapy, such as concurrent blockade of the inhibitory receptor NKG2A (Fig. 6d).

## Discussion

Despite unprecedented success in treating B-cell malignancies, many patients that receive CAR-T therapy ultimately relapse, highlighting the critical need to understand resistance mechanisms in order to improve clinical efficacy[5,8–13]. Taking a novel pool-based screening approach in a murine B-ALL model, we observed that in vivo loss of IFNγR/JAK/STAT signaling and components of the antigen processing and presentation pathway, rendered tumor cells more sensitive to CAR-T therapy. A large-scale follow up validation screen allowed us to hone in on $Qa\text{-}1^b$, the murine homolog of HLA-E and a downstream target of IFNγR signaling. In fact, B-ALL cells unable to signal through the IFNγR/JAK/STAT pathway were functionally deficient for $Qa\text{-}1^b$ suggesting that the mechanism by which IF IFNγR/JAK/STAT signaling promotes CAR-T resistance is, at least in part, dependent on IFNγ-induced $Qa\text{-}1^b$ surface expression on tumor cells. Furthermore, we saw that CAR-T efficacy was enhanced by blocking NKG2A, the inhibitory receptor of $Qa\text{-}1^b$ present on NK cells and subsets of T-cells. These data are consistent with a recent comprehensive in vivo screen in an immunocompetent mouse model of PDAC showing that $Qa\text{-}1^b$ suppression dramatically sensitizes tumors to ICB therapy[29].

Interestingly, NKG2A has been implicated as a novel immune checkpoint protein and studies have shown that blocking antibodies against it have exhibited antitumor effects by acting on endogenous NK and T-cell populations[54,55,59]. While these studies have focused on the application of this antibody alone, our data suggests that CAR-T efficacy could be improved by combination with an NKG2A blocking antibody. At present, the exact mechanism behind this benefit is unclear and interestingly, may not involve endogenous NK cells but instead the direct interaction of tumor cell $Qa\text{-}1^b$ and CAR-T expressed NKG2A, resulting in resistance to CAR-T therapy. In support of this hypothesis, we found that the addition of an NK1.1 blocking antibody, commonly used to deplete NK cells in vivo, significantly improved CAR-T efficacy leading to survival extension. A possible explanation for this result is that NK cells serve as an important source of IFNγ in the leukemia microenvironment allowing for increased tumor cell $Qa\text{-}1^b$ induction prior to and during CAR-T therapy, thus rendering the cells more resistant. In this setting, we suspect that IFNγ produced by CAR-T and endogenous immune cells acts as a double-edged sword in that it induces expression of $Qa\text{-}1^b$ (and potentially other inhibitory proteins) on tumor cells but also helps recruit and activate endogenous immunity that can both aid and abet CAR-T therapy. Experiments we conducted in immunocompetent versus immunodeficient mice transplanted with B-ALL and treated with CAR-T provide support for the beneficial effects of endogenous immune cell recruitment, with increased survival and decreased relapse seen in immunocompetent mice compared to immunodeficient animals. Other groups have

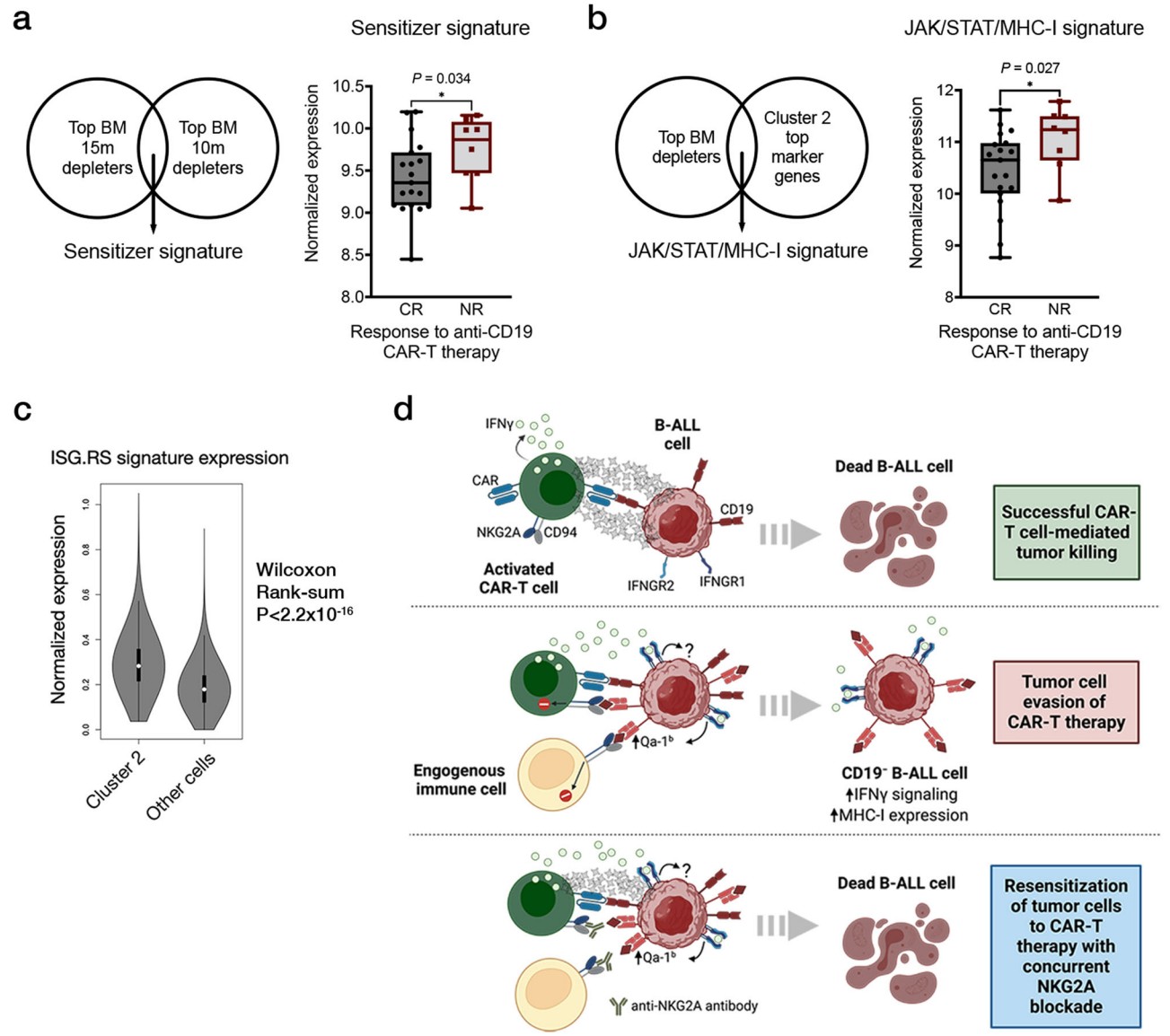

**Fig. 6 | JAK/STAT signaling is a potential therapeutic target in human B-ALL that can be exploited to enhance CAR-T therapy. a** Normalized expression of a sensitizer gene signature (composed of top depleting genes in the BM arms of our screen) in pre-treatment leukemia samples from patients who are complete responders (CR) or non-responders (NR) to anti-hCD19 CAR-T therapy. The five number summaries of the CR and NR boxplots are 8.45, 9.09, 9.36, 9.72, 10.20, and 9.05, 9.47, 9.87, 10.08, 10.15, respectively. **b** Normalized expression of a JAK/STAT/MHC-I resistance signature in the same pre-treatment leukemia samples shown in (a). The five number summaries of the CR and NR boxplots are 8.77, 10.01, 10.65, 10.98, 11.62, and 9.87, 10.64, 11.24, 11.50, 11.78, respectively. For (a-b), the plots are graphed from minima to maxima and all data points are overlaid. **c** Normalized expression of the immune checkpoint blockade resistance-associated interferon-stimulated genes (ISG.RS) signature in B-ALL cells (after CAR-T failure) residing in

expression cluster 2 or other cells. The ISG.RS signature is associated with poor outcomes in patients with large B-cell lymphoma treated with anti-CD19 CAR-T cells. A two-sided Wilcoxon rank-sum test was performed to compare the two distributions shown. An exact P value cannot be numerically determined due to ties in rank sum tests. A numerically approximate P value is shown. The plots are graphed in the Tukey method. The five-number summary for cluster 2 cells (left) is 0.036, 0.22, 0.28, 0.36, 0.57, and 0, 0.12, 0.18, 0.24, 0.42 for all other cells (right). **d** A final model for how high IFNγR/JAK/STAT signaling in tumor cells can promote resistance to CAR-T therapy via the upregulation of the NK and CD8+ T-cell inhibitory molecule Qa-1b, the murine homolog of HLA-E. Except where indicated, significance is determined using unpaired two-sided student's t-tests with Bonferroni correction for multiple comparisons. Data are mean ± s.e.m. *P < 0.05; **P < 0.01; ***P < 0.001; ****P < 0.0001.

reported similar effects. For example, a recent study in an immuno-competent murine model of GBM found that CAR-T cells induce endogenous immune cell activation necessary for full antitumor efficacy[27]. In addition, clinical studies in GBM have shown that CAR-T cells activate host immune cells[60,61]. We also found that combining CAR-T therapy with IFNγ blocking antibodies in vivo eliminated CAR-T efficacy in our model, potentially suggesting that IFNγ is important not just for CAR-T cell function but also activation and recruitment of endogenous immunity. On the other hand, NK cell depletion experiments showed significantly increased survival and CAR-T efficacy

highlighting the potential antagonistic effects of endogenous immunity.

Our examination of published patient data revealed that enhanced inflammatory signaling was characteristic of a CAR-T resistant microenvironment - patient tumor samples with elevated JAK/STAT/MHC-I expression prior to CAR-T treatment were associated with poor outcomes. Overall, these data paint a complex picture of the potential beneficial and harmful effects of IFNγ and inflammatory signaling in the context of CAR-T therapy. It is possible that the overall amount and timing of enhanced IFNγ and inflammatory signaling prior

to or during CAR-T therapy can influence its effect. Our data and published patient data suggest that enhanced IFNγ and inflammatory signaling prior to CAR-T administration can prime tumors for CAR-T resistance (possibly through induction of CAR-T inhibitory surface proteins like Qa-1[b]). In addition, in vitro we observe that PDAC cells only exhibit resistance to CAR-T therapy when pre-treated with IFNγ. We also observe related results in B-ALL cells whereby IFNγ pre-treatment can increase resistance to CAR-T therapy. Importantly, these B-ALL cells show significantly increased induction of Qa-1[b] after IFNγ pre-treatment which we believe is responsible, at least in part for the observed resistance. Ultimately, these data highlight a potential strategy to improve CAR-T efficacy by combining this cell therapy with an NKG2A blocking antibody.

Recent studies using human cell lines and xenograft mouse models suggest a complex context-dependent role for IFNγ. Data from one study suggested that IFNγ production by CAR-T cells was not required for CAR-T antitumor efficacy in xenograft models of B-ALL or lymphoma[23]. However, IFNγR signaling was important for susceptibility of solid tumors to CAR-T mediated killing, and the authors show that this was due to IFNγR-dependent upregulation of adhesion molecules (especially ICAM-1) in solid tumors[25]. In hematologic malignancies, there appear to be other adhesion molecules that stabilize the CAR-T/tumor interaction, namely the CD2/CD58 axis, that are not regulated by IFNgR signaling[62,63]. In this study, loss of IFNγR signaling emerged as a sensitizing pathway for CAR-T killing in an aggressive murine model of B-ALL, yet blockade of IFNγ cytokine by antibody did not enhance susceptibility to CAR-T cells, and, rather, resulted in abrogation of anti-tumor effects of CAR-T cells. We envision multiple factors contributing to these apparent discrepancies. First, our own study highlights how the role of IFNγ and IFNγR signaling can vary dramatically based on context. We find that in vitro, loss of IFNγR signaling promotes potent B-ALL tumor cell resistance to CAR-T killing while in vivo loss results in sensitivity. It has been proposed that CAR-T cells can kill tumor cells through both direct and indirect mechanisms, including IFNγ release[28]. It is possible, however, that depending on the specific context one mechanism may dominate[27]. Indeed, we find some evidence of both direct and indirect CAR-T mediated killing in our data. CD19 loss is enriched in all in vivo treatment contexts in our study, suggesting that CAR-T killing requires direct interaction with CD19-positive tumor cells in all tumor microenvironments. However, CD19 loss is not significantly enriched in cell culture treatment with CAR-T cells, implicating indirect IFNγ-mediated cell death as the primary mechanism of in vitro leukemia cell death. Additionally, we find that the impact of CD19 loss is significantly more dramatic in spleen versus bone marrow, and the loss of IFNγR potently sensitizes splenic B-ALL cells to CAR-T therapy but is much less potent as a sensitizer in bone marrow tumor cells. These data support a model by which CAR-T-mediated leukemia cell killing in the spleen is largely direct, whereas bone marrow leukemia cell killing is induced by direct CAR-T killing as well as perhaps some level of indirect killing via IFNγ release. Moreover, hypersensitization to IFNγ, as occurs in the absence of Ptpn2 or Fitm2, can alter the mechanism of in vivo CAR-T killing from direct to indirect.

Additional factors contributing to discrepancies in study findings include the ability of IFNγR signaling to induce immunomodulatory cell surface proteins on tumor cells or simply kill tumor cells. While our leukemia model effectively induces Qa-1[b] in response to IFNγR signaling, other tumor models may not. Induction of immunomodulatory cell surface proteins certainly varies between tumor types and potentially even between individuals[30,47]. In addition, different tumors, even within the same tumor type, can exhibit differential sensitivity to IFNγ with sensitive tumors dying at low levels of exposure and resistant ones incapable of being killed or requiring extremely high levels of exposure[28]. Thus, the effect of IFNγ on tumor cell response to CAR-T therapy will not only depend on context and timing but also the ability

of tumors cells to induce inhibitory receptors like HLA-E in response to inflammatory signals, as well as their general sensitivity to IFNγ-mediated death. Lastly, the impact of IFNγ signaling on targets other than tumor cells, for example, endogenous immune cells, likely also contributes to discrepancies between the in vitro and immunodeficient settings compared to immunocompetent settings. Our data, along with data from other studies implicates CAR-T induced endogenous immune activation, potentially through IFNγR signaling, in overall anti-tumor activity[29,30]. In addition, there are likely other complex interactions between the tumor microenvironment, CAR-T cells, and the tumor cells themselves that affect efficacy. In the in vitro and immunodeficient settings, one cannot observe the impact of these interactions as these settings are incapable of truly recapitulating complex tumor microenvironments. Thus, it is not unexpected that studies conducted in immunocompetent versus in in vitro or immunodeficient settings would yield different results. Ultimately, however, useful information can be derived from both settings, and we believe that combining insights from multiple models will be crucial for understanding clinically relevant mechanisms of CAR-T resistance and relapse.

Lastly, a number of groups have employed screening-based functional genomics approaches to interrogate tumor-intrinsic mechanisms of CAR-T killing. However, these have all been conducted in vitro[24,25]. Our screen is the first to be performed at genome scale in vivo. The generation of a pooled sgRNA library that was scaled according to tumor cell engraftment efficiency was critical to generate comprehensive in vivo screening data in which all sgRNAs could be represented. A key benefit of such pool-based library is its modularity, allowing for use in a broad range of in vivo tumor settings with distinct engraftment constraints. However, pool-based screens also require a subsequent validation step to directly compare scoring sgRNAs from distinct component pools. While this approach adds significant additional challenges compared to whole-genome in vitro approaches, given the complexity of the tumor microenvironment and the interplay between endogenous immunity and immunotherapies like CAR-T treatment, conducting these screens in vivo can provide valuable insights to improve therapies that would not have been uncovered otherwise.

## Methods
### Pooled sgRNA screening
A custom genome-wide library divided into 48 sub-pools was generated in collaboration with the Broad Institute's Genetic Perturbation Platform (GPP). In total, 97,336 unique guides targeting the protein-coding regions of 21,958 unique murine genes with 4 sgRNAs each (plus control non-targeting and intragenic cutting guides) were included. All protein-coding murine genes were subdivided into 48 pools by their initial KEGG term (obtained using KEGG REST API in BioPython, biopython.org), in a non-redundant manner. All four guides targeting the protein-coding region of any given gene were kept together in the same pool. Using this approach, only 36% of protein-coding genes could be classified into a KEGG pathway. Thus, the first 14 sub-pools and part of sub-pool 15 were filled by KEGG genes. All other genes were randomly distributed among the remaining sub-pools. Mouse essential genes (defined as orthologous mouse genes for the human essential gene set from Hart et al., 2015, 1530 genes; obtained from Ensembl Biomart) were divided evenly across all sub-pools. Guides against human EGFRvIII, human CD22, human CD19, and murine Cd19 were included in the first sub-pool[64]. Guides against olfactory genes (1133 total) were also distributed evenly amongst all sub-pools. All 48 sub-pools were cloned into a lentiviral pRDA-Crimson_170 vector (Fig. 1o). To preserve library complexity, a minimum of 1000-fold coverage of the sgRNA library was maintained at each in vitro step before the screen, and at a minimum of 150-fold coverage (range: 153 to 203-fold coverage in vivo, all in vitro screens were performed above

500x) was maintained in all screens completed. Pool A consisted of sub-pools 1-8 and had a total of 15,308 sgRNAs targeting the protein-coding regions of 3648 unique mouse genes. Pool B consisted of sub-pools 9-16 and had a total of 15,147 sgRNAs targeting the protein-coding regions of 3648 unique mouse genes. Pool C consisted of sub-pools 17-24 and had a total of 15,258 sgRNAs targeting the protein-coding regions of 3648 unique mouse genes. Pool D consisted of sub-pools 25-32 and had a total of 17,335 sgRNAs targeting the protein-coding regions of 3648 unique mouse genes. Pool E consisted of sub-pools 33-40 and had a total of 14,713 sgRNAs targeting the protein-coding regions of 3648 unique mouse genes. Pool F consisted of sub-pools 41-48 and had a total of 19,575 sgRNAs targeting the protein-coding regions of 3718 unique mouse genes. Cloned and sequenced plasmid pools, and viral supernatant were generated by the Broad Institute's GPP.

For screens, Cas9$^+$ cells were thawed, recovered, and expanded for 5 days to ensure robust growth, and then tested for cutting efficiency to ensure high rates of editing efficiency[35,65]. After cutting assays were completed, cells were expanded over three additional days and infected with sub-pools. For each of the 48 sub-pools, $6 \times 10^7$ cells were spin-infected with predetermined amounts of viral supernatant such that 15-30% of all cells were infected (expressed E2-Crimson, and survived puromycin selection; MOI«1). The resulting cells were resuspended at a concentration of $10^6$ cells/mL in virus-containing medium supplemented with 10 µg/mL polybrene (Sigma), divided into 6-well plates, and centrifuged at 1000 x $g$ and 37 °C for 1.5 hrs. Cells were then pooled into flasks and cultured overnight. Thirty-six hours later, cell density was adjusted to $10^6$ cells/mL (and was never allowed to go over $3 \times 10^6$ cells/mL) and puromycin selection (2 µg/mL, Gibco, A1113803) was started. Cells were selected over two days and then spun out of puromycin-containing medium. Cells were then allowed to recover for one day, after which the appropriate number of infected, selected, and recovered cells were flow cytometry sorted and combined into their respective pools. The next day, cells from the large, combined pools were prepared for tail vein injection into mice or kept in culture for the in vitro screens. Two days later, CAR-T cells were adoptively transplanted into mice via tail vein injection at the indicated doses. For all in vitro screens, $1.4 \times 10^7$ library cells were seeded and treated two days later (on the same schedule as mice) with control CAR-T cells (anti-hEGFRvIII), anti-mCD19 CAR-T cells, or with no CAR-T cells, at the E:T ratios indicated in the text. In vitro CAR-T screens were set up in triplicate while the no CAR-T condition was kept in a single plate. Input samples were collected just after puromycin selection had been completed (Input PS) and on the day cells were injected into mice/set up for in vitro screens (Input or Input DOI). Upon becoming moribund, mice were sacrificed and E2-Crimson$^+$ cells were sorted from the bone marrow and spleen (average number sorted cells per compartment = $1.38 \times 10^7$). For in vitro screens where the B-ALL cells viability was above 95% and represented more than 99% of all cells present in the sample, cells were counted and $2 \times 10^7$ cells were collected for gDNA isolation. The only condition that did not meet this cutoff was the anti-mCD19 at E:T of 1:2 for all 6 pools. Here, CAR-T cells were completely gone from culture, but B-ALL viability of cells remained low at 30–60% across replicates, necessitating sorting to isolate cells ($1.4–2.0 \times 10^7$ sorted per replicate, per condition). For all flow cytometry experiments in animal samples (screens and validation), data for a minimum of 10,000 live cells (via DAPI exclusion) were collected and analyzed.

Finally, gDNA from all cells were isolated using the Machery Nagel L Midi NucleoSpin Blood Kit (Clontech, 740954.20). Modifications to the manufacturer's instructions were added as follows: in step 1, cells were lysed in the kit's proteinase K containing lysis buffer for longer (overnight at 70 °C). The next morning, lysates were allowed to cool to room temperature, 4.1 µL of RNase A (20 mg/mL; Clontech, 740505)

was added, and cells were incubated for 5 minutes at room temperature. The procedure then continued as indicated by the manufacturer. PCR inhibitors were removed from the resulting gDNA (Zymo Research, D6030) and the concentration of the resulting gDNA was measured using the Qubit dsDNA HS assay kit (ThermoFisher, Q32854), and if necessary, diluted to 200 ng/µL with elution buffer. gDNA was then submitted for Illumina sequencing. Data were analyzed as specified in the Data Analysis section of this manuscript.

## Bulk transcriptome profiling
Total RNA was extracted from $1 \times 10^6$ cells per sample using the Macherey-Nagel Nucleospin RNA Plus kit, and RNA sample quantity and quality was confirmed using an Agilent Fragment Analyzer. RNA-seq libraries were created from 250 ng of total RNA using the NEBNext UltraII Directional RNA Library Prep kit (New England Biolabs) using half volume reagents and 14 cycles of PCR. Illumina library quality was confirmed using the Fragment Analyzer and qPCR and sequenced on an Illumina NextSeq500 using v2.1 chemistry with 40nt paired end reads (RTA version 2.11.3). Data were analyzed as specified in the Data Analysis section of this manuscript.

## Single cell transcriptome profiling
Mice were transplanted with $3 \times 10^6$ B-ALL cells and challenged with $10^7$ CAR-T cells targeting either mCD19 or a control epitope (human EGFRvIII). Upon becoming moribund, mice were sacrificed, and B-ALL cells were sorted by FACS from the bone marrow and spleen. For each sample, approximately 10,000 live B-ALL cells (via DAPI exclusion) were sorted for transcriptional profiling. Single-cell expression libraries were prepared using the 10x Genomics Chromium v3 reagents. Data were analyzed as specified in the Data Analysis section of this manuscript.

## Mouse maintenance
All mouse experiments were conducted under Institutional Animal Care and Use Committee (IUCAC)-approved animal protocols at the Massachusetts Institute of Technology (MIT). The mouse strains used in this study included either male and female C57BL/6 (The Jackson Laboratories) and either male or female NOD/SCID/IL2Rg – /– (NSG; The Jackson Laboratories) mice, as indicated for each in the mouse studies methods section of the manuscript. All experimental mice used were 8-12 weeks old. Mice were housed under social conditions (two to five mice per cage) on a 12-hour dark/12-hour light cycle, ambient temperature 21 °C ± 1 °C, and humidity 50% ± 10%. All animals were housed in the pathogen-free animal facility of the MIT Koch Institute, in accordance with the animal care standards of the institutions. Food and water was provided ad libitum. All animal research at MIT is conducted under humane conditions with utmost regard for animal welfare. The animal care facility staff is headed by a chief veterinarian and includes a veterinary assistant, animal care technicians, and administrative support. All facility staff are members of the American Association of Laboratory Animal Science. MIT adheres to institutional standards for the humane use and care of animals, which have been established to assure compliance with all applicable federal and state regulations for the purchase, transportation, housing, and research use of animals.

## Mouse studies
All mouse experiments were approved by the MIT Committee on Animal Care (CAC) and the Department of Comparative Medicine (DCM). For leukemia experiments, mice were monitored for disseminated tumor burden by overall body condition score. All mice were euthanized upon the appearance of morbidity, in accordance with CAC and DCM policy. Immunocompetent recipient C57BL/6 male mice were sub-lethally irradiated (1 × 5 Gy) immediately prior to transplantation with B-ALL cells and underwent ACT of CAR-T cells

2 days later, as noted in the text. For in vivo screens, male mice were injected with $3 \times 10^6$ Cas9$^+$ library B-ALL cells and the indicated number of CAR-T cells. Both B-ALL and CAR-T cells were prepared for transplantation by resuspending in 200 μL Hank's balanced salt solution (Sigma-Aldrich, Catalog# H9394) and loaded in 27.5-gauge syringes (Becton Dickinson, Catalog# BD 305620). All cell solutions were administered via tail vein injections.

GBM mice were monitored for any signs of disease and were sacrificed at low tumor burden, prior to developing morbidity, in accordance with CAC and DCM policy. For validation experiments conducted with the murine glioblastoma line, $0.5 \times 10^6$ Gl261 cells were delivered via intracranial injection into female C57BL/6 mice. Mice were then treated four days later with an intracranial injection of $0.2 \times 10^6$ CAR-T cells. Surgical procedure closely followed that of previous studies using this model conducted in our lab[66].

For in vivo blocking of IFNγ, mice were injected intraperitoneally (i.p.) every third day with 200 μg of *InVivo*MAb anti-mouse IFNγ antibody (Bio X Cell, clone XMG1.2, Catalog# BE0055) or 200 μg of *InVivo*MAb anti-horseradish peroxidase control antibody (clone HRPN, Bio X Cell, Catalog# BE0088), starting the day after ACT.

For in vivo inhibition of JAK1/2, Ruxolitinib (Selleckchem, INCB018424) was resuspended according to manufacturer guidelines and mice were dosed every 12 hours via oral gavage with 90 mg/kg Ruxolitinib or vehicle control, starting the day after disease transplantation and continuing until moribund.

For in vivo blocking of NK1.1, mice were injected i.p. on the same day or one day prior to ACT, and then once again 7 days later, with 200 μg of InVivoMAb anti-mouse NK1.1 antibody (clone PK136, Bio X Cell, Catalog# BE0036) or 200 μg of *InVivo*MAb mouse IgG2a isotype control antibody (clone C1.18.4, Bio X Cell, Catalog# BE0085).

For in vivo blocking of NKG2A, mice were tail vein injected with 200 μg of InVivoMAb anti-mouse NKG2A antibody (clone 20D5, Bio X Cell, Catalog# BE0321) or 200 μg of InVivoMAb rat IgG2a isotype control antibody against trinitrophenol (clone 2A3, Bio X Cell, Catalog# BE0089) on the same day as ACT. Mice were re-dosed with their respective antibody three and six days later and monitored for survival.

For in vivo glioma competitive assays, $0.5 \times 10^6$ GL261 cells composed of a 50:50 mix of WT and *Ifngr1* KO cells were delivered via intracranial injection into female C57BL/6 mice. Mice were then treated four days later with an intracranial injection of $0.2 \times 10^6$ anti-hCD19 murine or control CAR-T cells. Tumors were harvested four days after treatment.

### Bioluminescence studies
XenoLight D-Luciferin Potassium Salt D (PerkinElmer, Catalog# 122799) was used for standard bioluminescent imaging (resuspended at 30 mg/mL in saline, sterile filtered, and stored at −80 °C). Mice were weighed and luciferin was loaded in 27.5-gauge syringes and administered via intraperitoneal injection at a dose of 165 mg/kg. Mice were then anesthetized with 2.5% isoflurane (Piramal Critical Care, NDC# 66794-013-25), delivered at 1 L per minute in O$_2$. Ten minutes from the time of luciferin injection, animals were imaged on a Xenogen IVIS system at consistent exposures between groups with small binning. Data was analyzed using Living Image version 4.4 software (Caliper Life Sciences). Images were normalized to the same color scale for figure generation.

### Cell culture
All cell lines were mycoplasma negative.

**Murine B-ALL cells.** Cells were cultured in RPMI with L-glutamine (Corning, 10-040-CM), supplemented with 10% fetal bovine serum (FBS) and 2-mercaptoethanol to a final concentration of 0.05 mM (Gibco, 21985023).

**Murine B-cell lymphoma cells (Eμ-Myc).** Cells were cultured in medium composed of a 50:50 mix of IMDM with L-glutamine and 25 mM HEPES (Corning, 10-016-CM) and DMEM with L-glutamine and sodium pyruvate (Corning, 10-013-CM), supplemented with 10% FBS and 2-mercaptoethanol to a final concentration of 0.05 mM (Gibco, 21985023).

**Murine Glioblastoma cells (Gl261).** Cells were cultured in DMEM with L-glutamine and sodium pyruvate (Corning, 10-013-CM) supplemented with 10% FBS.

**Murine pancreas cancer cells (KPC).** ***

**Murine T cells.** T-cells harvested from the spleens of mice were cultured in plates coated with activating antibodies (as described in CAR-T cell production methods) in T-cell medium (TCM): RPMI with L-glutamine (Corning, 10-040-CM), supplemented with 10% FBS, recombinant human IL-2 (rhIL-2, final concentration of 20 ng/mL; Peprotech, Cat# 200-02-1 mg), and 2-mercaptoethanol to a final concentration of 0.05 mM (Gibco, 21985023).

**Human cell lines.** HEK293T (293 T) cells were cultured in DMEM with L-glutamine and sodium pyruvate (Corning, 10-013-CM) supplemented with 10% FBS. Raji cells were cultured in RPMI with L-glutamine (Corning, 10-040-CM) supplemented with 10% FBS.

### Viral supernatant production
Viral supernatant was produced using standard methods. Briefly, 293 T cells were transfected with retroviral or lentiviral transfer plasmid and packaging vector (retrovirus: pCL-Eco, Addgene, 12371; lentivirus: psPAX2, Addgene, 12260 with VSVg envelop plasmid pMD2.G, Addgene, 12259) using Mirus TransIT-LT1 (Mirus, MIR2305) as indicated by the manufacturer. The next day, 293T cells were switched into medium composed of 60% RPMI complete and 40% DMEM complete. Viral supernatant was collected 24 and 48 hours after transfection, passed through a 0.45μm filter to remove residual 293 T cells, and stored at 4 °C for a maximum of four days.

### In vitro killing assay
In vitro CAR-T killing assays were performed using standard methods[67,68]. Briefly, target cells are counted and co-cultured with or without CAR-T cells at indicated E:T ratios (accounting for CAR-T infection rate) in RPMI complete, supplemented with 2-mercaptoethanol and 10% FBS, but no rhIL-2. Sixteen to twenty-four hours later, the total cell number per well was counted and the cell suspension was analyzed by flow cytometry to assess for live/dead (via DAPI stain), %mCD19$^+$ cells, and %CD8$^+$ cells. The densities of each cell type (CAR-T, target cell, non-transduced T-cell) were also determined via flow cytometry. The resulting target cell densities in CAR-T containing wells are then normalized to the resulting target cell density in control wells seeded with the same number of target cells but without CAR-T cells. For all flow cytometry experiments, data for a minimum of 10,000 live cells (via DAPI exclusion) were collected and analyzed.

### Interferon gamma ELISA release assay
Standard methods were used for the enzyme-linked immunosorbent assay (ELISA). Briefly, supernatant from in vitro CAR-T killing assays was collected and centrifuged to remove any contaminating cells. IFNγ released into the supernatant by CAR-T cells was then measured using the DuoSet ELISA kit for mouse IFNγ (R&D systems, DY485) and Nunc MaxiSorp flat bottom plates (Thermo Fisher Scientific, 44-2404-21) on a Tecan infinite 200 Pro machine, as indicated by the manufacturer. To ensure that the assay was completed within the linear range of the kit, supernatant is initially diluted 1:10 in reagent diluent. At least six serial

4-fold dilutions were then performed. At least one standard curve for this assay was generated per plate and at least two standard curves for the entire experiment were constructed using standard solutions supplied by the manufacturer. The substrate solution was 1-Step™ Ultra TMB-ELISA (Thermo Fisher Scientific, 34028) and the stop solution was 2 N sulfuric acid (VWR, BDH7500-1). Bovine serum albumin (BSA; Sigma, A8022-500G) was prepared as a sterile filtered 5% stock in PBS (Corning, 21-031-CV).

### CAR-T cell production

Before collecting T-cells, 6-well plates were coated overnight with activating antibodies against mCD3e (Bio X-Cell, BE0001-1) and mCD28 (Bio X-Cell, BE0015-1) at 5 µg/mL each in PBS (Corning, 21-031-CV) at 4 °C. The next day, 8–12-week-old male C57BL/6 mice (Jackson) were sacrificed, and their spleens were collected. $CD8^+$ T-cells were isolated using Miltenyi Biotec CD8a (Ly-2) MicroBeads for mouse (positive selection kit; Miltenyi, 130-117-044) and LS columns (Miltenyi, Cat# 130-042-401) as indicated by the manufacturer. Coated plates were rinsed once with PBS and T-cells were resuspended at $0.5 \times 10^6$ to $10^6$ cells/mL in T-cell medium (TCM, recipe in cell culture methods). After 24 hours, activated T-cells were collected and placed into fresh TCM after counting. Cell concentrations were then adjusted to $10^6$ cells/mL in a 50:50 mix of TCM:retroviral (RV) supernatant supplemented with protamine sulfate to a final concentration of 10 µg/mL (MS Biomedicals, ICN19472910), 2-mercaptoethanol to a final concentration of 0.05 mM, and rhIL-2 to a final concentration of 20 ng/mL. Once resuspended, cells were spin-infected at 1000xg for 1.5 hrs at 37 °C on new antibody-coated plates. The next day, T-cells were collected from plates, resuspended in fresh TCM at a cell density of $0.5 \times 10^6$ to $10^6$ cells/mL, and re-plated on new antibody-coated plates. Twenty-four hours later, T-cells were collected from antibody-coated plates and prepared for tail vein injection into animals or for in vitro kill assays/screens, as described above. T-cells were always cultured and infected on PBS rinsed, antibody-coated 6-well plates, as described above, except during in vitro killing assays/screens where no activating antibodies were ever used.

### Western blotting

Cells were lysed with RIPA buffer (Boston BioProducts, BP-115) supplemented with 1X protease inhibitor mix (cOmplete EDTA-free, 11873580001, Roche). Protein concentration of cell lysates was determined using Pierce BCA Protein Assay (ThermoFisher Scientific, 23225). Total protein (40-60 µg) was separated on 4-12% Bis-Tris gradient SDS-PAGE gels (Life Technologies) and then transferred to PVDF membranes (IPVH00010, EMD Millipore) for blotting.

### Plasmids, cloning, and sgRNAs

**Packaging and envelope plasmids used for viral production.** Retrovirus: pCL-Eco (Addgene, 12371)
Lentivirus: psPAX2 (Addgene, 12260) with VSVg envelop plasmid pMD2.G (Addgene, 12259) or pCMV-EcoEnv (Addgene, 15802)

**Chimeric antigen receptor (CAR) plasmids.** The murine CD19-targeting second-generation CAR 1D3-28Z.1-3, containing inactivating mutations in the $1^{st}$ and $3^{rd}$ ITAM regions of the CD3-ζ chain, was synthesized by Twist Bioscience and cloned into the GFP$^+$ MP71 retroviral vector[36,69]. The clinically used scFv sequence (heavy chain linked to light chain variable regions) against human CD19, FMC63, was provided by the Maus lab. A CD28-containing $2^{nd}$ generation murine CAR-Targeting hCD19 protein was then constructed by switching out the scFv for 1D3-28Z.1-3 in the anti-mCD19 CAR and replacing it with the FMC63 scFv (Twist Bioscience). The same technique was used for the 3C10 scFv targeting human EGFRvIII, which was reported by the Rosenberg lab[70]. All CAR constructs are identical, containing a CD8a

leader sequence, followed by the respective scFv, followed by an IgG4 hinge sequence, a portion of the murine CD28 molecule from amino acids IEFMY to the 3′ terminus, and finally, the cytoplasmic region of the murine CD3-ζ chain from amino acids RAKFS to the 3′ terminus with the tyrosine in both ITAMs 1 and 3 mutated to phenylalanine as described (all in frame)[36]. All CAR constructs were extensively tested to ensure that they only targeted their peptide of interest. Human EGFRvIII expression was induced using pMSCV-XZ066-EGFRvIII (Addgene, plasmid 20737) and murine hEGFRvIII$^+$ B-ALL cells were generated. Retroviral supernatant to induce hCD19 expression was provided by the Maus lab. All CARs were extensively tested in vitro (killing assays methods) and in vivo (as described in text) to ensure no off-target effects were present.

**CRISPR plasmids.** To generate Cas9$^+$ murine B-ALL cell lines, lentiCas9-Blast (Addgene, 52962) was used, and cells were selected with Blasticidin (Gibco, A1113903) at 20 µg/mL for seven days, single cell cloned via FACS, and then assayed for Cas9 expression via WB. All guide RNAs were designed using the Broad Institute's sgRNA Designer tool (https://portals.broadinstitute.org/gppx/crispick/public) and cloned into lentiGuide-Puro (Addgene, 52963) for the functional cut assay (tracking loss of mCD19 on the cell surface) or pRDA-Crimson_170 to generate KOs of indicated genes[35]. Guide RNAs used are below (Forward/Reverse):

Murine *Cd19* sgRNA#1: 5′-GAATGACTGACCCCGCCAGG-3′
Murine *Cd19* sgRNA#2: 5′-GCAATGTCTCAGACCATATGG-3′
Murine *Ifngr1* sgRNA#1: 5′-GGCTCGGAGAGATTACCCGA-3′
Murine *Ifngr1* sgRNA#2: 5′- GTATGTGGAGCATAACCGGAG-3′
Murine *Stat1* sgRNA#1: 5′-GACCCCAGTCTTCAAGACCAG-3′
Murine *Stat1* sgRNA#2: 5′-GTGTGATGTTAGATAAACAGA-3′
Murine *Jak2* sgRNA#1: 5′-CACGGGACACTCCGTATCTGC-3′
Murine *Jak2* sgRNA#2: 5′-GCAGATACGGAGTGTCCCGTG-3′
Murine *H2-T23* sgRNA: 5′-GTACTACAATCAGAGTAACGA-3′
Murine *Ptpn2* sgRNA: 5′- AAGAAGTTACATCTTAACAC -3′
Murine *Fitm2* sgRNA: 5′- CCCGATGCACTCACACGTTG -3′
Neutral guide sgRNA: 5′-GACAACCCCAACCCCGATACT-3′
*lacZ* sgRNA: 5′-GTGCGAATACGCCCACGCGAT-3′

**Other plasmids.** MSCV-mCherry (Addgene, 52114) was used to generate 20.12DP cells from mCherry$^-$ GFP$^+$ 20.12 cells.

### Generating Qa-1$^b$ knockout B-ALL cells

B-ALL cell lines deficient for *Qa-1$^b$* were generated using CRISPR-Cas9 technology. Single guide RNAs directed against the *H2-T23* locus were designed and cloned into pRDA-Crimson_170, as described above. Cas9$^+$ B-ALL cells (clone RH62) were transduced with sgRNAs and selected with puromycin over 48 hours. Loss of *Qa-1$^b$* was confirmed by incubating transduced cells with 30 IU/mL IFNγ for 24 hours and subsequently analyzing Qa-1$^b$ surface expression by flow cytometry, as previously described[54]. Cells that were negative for Qa-1$^b$ were then FACS sorted twice until a pure population of *H2-T23$^{-/-}$* cells was established.

### Antibodies

**Western blotting.** anti-β-ACTIN (Cell signaling, catalog# 4967 S), anti-CD19 (Abcam, catalog# ab25232), anti-Cas9 (ActiveMotif, catalog# 61577), anti-JAK2 (Cell Signaling Technology, catalog# 3230), anti-IFNγR1/CD119 (R&D Systems, catalog# MAB10261), anti-STAT1 (Cell Signaling Technology, catalog# 9172), anti-rabbit IgG HRP-linked antibody (Cell Signaling Technology, catalog# 7074), anti-mouse IgG HRP-linked antibody (Cell Signaling Technology, catalog# 7076), anti-rat IgG HRP-linked antibody (Cell Signaling Technology, catalog# 7077), Rabbit anti-Armenian Hamster IgG H&L (HRP) (Abcam, catalog# ab5745).

**Flow cytometry.** anti-mouse CD19-BV785 (BioLegend, catalog# 115543), anti-mouse CD8-PE/Cy7 (BioLegend, catalog# 100722), anti-mouse CD8-GFP (BioLegend, catalog# 100706), anti-human CD19-APC/Cy7 (BioLegend, catalog# 302218), anti-mouse Qa-1$^b$-BV786 (BD Biosciences, catalog# 744390), anti-mouse Nk1.1-BV785 (BioLegend, catalog# 108749).

**In vivo blocking antibodies.** *InVivo*MAb anti-mouse IFNγ antibody (Clone XMG1.2, Bio X Cell, Catalog# BE0055), *InVivo*MAb anti-horseradish peroxidase control antibody (clone HRPN, Bio X Cell, Catalog# BE0088), InVivoMAb anti-mouse NKG2A antibody (clone 20D5, Bio X Cell, Catalog# BE0321), InVivoMAb rat IgG2a isotype control antibody against trinitrophenol (clone 2A3, Bio X Cell, Catalog# BE0089), *InVivo*MAb anti-mouse NK1.1 (Clone PK136, BioXCell, catalog# BE0036), *InVivo*MAb mouse IgG2a isotype control, unknown specificity (Clone C1.18.4, BioXCell, Catalog# BE0085).

### Data Analysis

**Screen hit discovery.** A series of 91 tab-delimited text files containing sample count data assigned to SKY library barcodes were produced by the Broad Institute's Genomic Perturbation Platform (GPP) using PoolQ (version 2.2.0) from three separate sequencing runs. These data were imported and organized into 12 dataframes, one for each pool of barcodes in both the invitro and in vivo experiments. This processing was done using R version 4.2.1 (R Core Team 2021), tidyverse 1.3.2. These dataframes were then used as input for differential comparison with MAGeCK test (version 0.5.9.4;) using adjust-method fdr, gene-lfc-method median and norm-method control using a list of negative control guides that target intergenic or olfactory gene sequences. Each pool included approximately 1000 negative control guides. After testing, guide and gene-level data results for each experiment and pool were imported, processed, and visualized in R using MAGeCKFlute (version 2.0.0) and tidyverse 1.3.2. For whole-genome consideration, pool data was aggregated into a single dataframe. The associations between guides and genes was done using assembled pool-level chip files. The full SKY library contains 88793 barcodes. Of these 82372 target a single gene based on our working annotation file. Barcodes that target multiple genes and genes targeted by less than 3 barcodes were deprioritized using a whitelist annotation file. Log fold change and FDR thresholds were then applied to prioritize enriched and depleted genes. All input count files, Rmd code, MAGeCK scripts and annotation files will be provided upon request.

**Bulk transcriptome analysis.** Paired-end RNA-seq data was used to quantify transcripts from the mm10 mouse assembly with the Ensembl version 101 annotation using Salmon version 1.3.0[71]. Gene level summaries were prepared using tximport version 1.18.0[72] running under R version 4.0.3.

**Single cell transcriptome profiling, data processing.** Sequencing data was aligned to the mm10 reference genome and converted to fastq files using bcl2fastq (v2.20.0.422). Cell count matrices were generated using cellranger (v.5.0). Matrices were analyzed by Seurat (v4.0.4) for R (v4.0.2). Digital gene expression matrices were filtered to exclude low-quality cells (<1000 UMI, <400 genes or > 8000 genes, > 50% mitochondrial reads). Low-quality cells were further filtered from the dataset using the variance sink method as previously described[73]. Briefly, data was normalized and scaled, and known cell cycle genes were regressed out[74]. Principal component analysis was performed on regressed and scaled data. The standard deviation of principal components was quantified using an elbow plot, and input dimensions for SNN clustering (EGFRv3 bone marrow = 35, EGFRv3 spleen = 42, CD19 bone marrow = 30, CD19 spleen = 30) at which standard deviation = 2. SNN clustering was performed to generate UMAP plots (k.param = 40, res = 0.5). Clusters containing

low-quality cells (50% of cells with > 10% mitochondrial reads) were removed from the dataset. After filtering, samples from bone marrow and spleen treated with EGFRv3-CAR-T or CD19-CAR-T were merged into a single dataset. Cell cycle phase was assigned using the cell cycle scoring function based on expression of known cell cycle genes[74]. Merged data set was normalized and scaled, cell cycle genes were regressed out. Principal component analysis was performed, and standard deviation of principal components was quantified by elbow plot. Nearest neighbors were found (dim = 50, k.param = 40) then clustered using SNN clustering (res = 0.5). Enriched genes for each cluster were identified with the cluster marker function. Cluster occupancy was quantified for each treatment condition and phase of the cell cycle to further define therapeutic response.

Manual analysis of flow cytometric data was carried out using FlowJo software (TreeStar).

In vivo bioluminescence data was analyzed using Living Image version 4.4 software (Caliper Life Sciences).

All statistical analyses outside of RNAseq, single cell sequencing, and screen hit analysis, were performed with GraphPad Prism 10 (GraphPad Software). The specific statistical tests performed are specified in figure legends. Differences are considered significant for $P$-values $\leq 0.05$, or as indicated in the manuscript text when adjustments for multiple hypothesis testing was required.

### Reporting summary

Further information on research design is available in the Nature Portfolio Reporting Summary linked to this article.

### Data availability

The RNAseq datasets generated from the BCR-ABL+ murine B-ALL model and analyzed during the current study are available in the GEO repository under the publicly available accession number GSE196143. The in vivo screening datasets generated and analyzed during the current study are available from the corresponding author on reasonable request. The in vivo screening datasets and analysis methodology, including the input count files, Rmd code, MAGeCK scripts, MAGeCK output and various annotation files for both the primary and validation screens are available in this github repository: https://github.com/KochInstitute-Bioinformatics/Ramos_Koch_Leuk. Source data for the validation screen are also provided with this paper. Source data are provided with this paper.

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

## Acknowledgements

The authors thank Hojun Li, and members of the Hemann and Vander Heiden labs for valuable discussions and intellectual input, the Koch Institute's Robert A. Swanson (1969) Biotechnology Center for technical support, specifically the Flow Cytometry and Preclinical Imaging and Testing facilities, and S. Levine from the MIT BioMicro Center for informative discussions about guide library and scRNA sequencing. The authors also thank Beatrice Grauman-Boss for her support generating CAR virus. This work was generously supported by the MIT Center for Precision Cancer Medicine, the Ludwig Center at MIT, a Margaret A. Cunningham Immune Mechanisms in Cancer Research Fellowship, a David H. Koch Graduate Fellowship, and the Paul and Daisy Soros Fellowship for New Americans. This work was also supported in part by NCI R01-CA233477, R01-CA226898, and NIH/NIAID R21AI151827 to M.T. Hemann and the Koch Institute Support (core) Grant P30-CA14051 from the NCI, and award Number T32GM007753 and T32GM144273 from the National Institute of General Medical Sciences. CAW was partially supported by Cancer Center Support (core) Grant P30-CA14051 from the NCI to the Barbara K. Ostrom (1978) Bioinformatics and Computing Core Facility. The content is solely the responsibility of the authors and does not necessarily represent the official views of the National Institute of General Medical Sciences or the National Institutes of Health. Figures 1n, 1q, 3a, 4a, 4e, 6d and Supplementary Fig. 5d were created with BioRender.com.

## Author contributions

A.R., Y.L.-L., C.K. and M.T.H. designed the study. A.R., A.M., Y.L.-L, C..K. R.H., K.A., J.F., D.G., K.E., K.G., K.Y., J.D. and P.L. conducted experiments. C.A.W. provided bioinformatic analysis. R.H. provided support with characterization of clone 20.12 and clone RH62, and the entirety of the primary screens. T.K. provided critical support in designing the CAR constructs and cloned all of them. YL analyzed scRNAseq data. RCL provided vital reagents. A.R. and K.A. completed the validation screen. AV, JGD, MVM, MGVH, RTM, and MEB gave vital advice and provided reagents. A.R., Y.L.-L, C.K., and M.T.H. wrote the manuscript. All authors reviewed and edited the manuscript.

## Competing interests

M.V.M. is on the Board of Directors of 2SeventyBio, and holds equity in TCR2, Century Therapeutics, Oncternal, and Neximmune, and has served as a consultant for multiple companies involved in cell therapies. R.T.M. has received consulting or speaking fees from Bristol Myers Squibb, Gilead Sciences and Immunai Therapeutics, has equity ownership in OncoRev, and receives research funding from Calico Life Sciences. M.G.V.H. declares he is an advisory board member for Agios Pharmaceuticals, Aeglea Biotherapeutics, Faeth Therapeutics, Drioa Ventures and iTeos Therapeutics, and a co-founder of Auron Therapeutics. M.E.B. is an equity holder in 3 T Biosciences, and is a co-founder, equity holder, and consultant of Kelonia Therapeutics and Abata Therapeutics. AR is a co-founder and equity holder of Celsius Therapeutics, an equity holder in Immunitas and, until 31 July 2020, was a scientific advisory board member of Thermo Fisher Scientific, Syros Pharmaceuticals, Neogene Therapeutics and Asimov. AR is an employee of Genentech from August 1st, 2020. The remaining authors declare no competing interests.

## Additional information

[1]Koch Institute for Integrative Cancer Research, Massachusetts Institute of Technology, Cambridge, MA, USA. [2]Department of Biology, Massachusetts Institute of Technology, Cambridge, MA, USA. [3]Department of Biological Engineering, Massachusetts Institute of Technology, Cambridge, MA, USA. [4]Cellular Immunotherapy Program, Cancer Center, Department of Medicine, Massachusetts General Hospital, Boston, MA, USA. [5]Immunology Program, Harvard Medical School, Boston, MA, USA. [6]Center for Cancer Research, Massachusetts General Hospital, Charlestown, MA, USA. [7]Broad Institute of Harvard and Massachusetts Institute of Technology, Cambridge, Massachusetts 02142, USA. [8]Klarman Cell Observatory, Broad Institute of MIT and Harvard, Cambridge, MA, USA. [9]Howard Hughes Medical Institute, Chevy Chase, MD, USA. [10]Dana-Farber Cancer Institute, Boston, MA, USA. [11]Ragon Institute of MIT, MGH, and Harvard, Cambridge, MA, USA. [12]Present address: Solid Tumors Program, Division of Oncology, Center for Applied Medical Research (CIMA), University of Navarra, Pamplona, Spain. [13]These authors contributed equally: Azucena Ramos, Catherine E. Koch, Yunpeng Liu. ✉e-mail: hemann@mit.edu

