## [Peer Review File · Nature Communications]

REVIEWERS' COMMENTS

Reviewer #1 (Remarks to the Author):

The authors additions and extended discussion are appreciated. My remaining concern would be the GL261 model.

In my opinion the inability to confirm these findings robustly in a second model is of concern. The authors reference new data regarding HLA-E in a pancreatic model. Whilst this aids further support to the concept of targeting this protein to enhance CAR T therapy, it of course does not support the concept that IFN γ is overall immunosuppressive because there are obviously many of IFN γ -responsive genes that modulate the response.

However, the authors discussion place these results into the overall context of the field and I do not believe this aspect alone should preclude publication of the results.

Minor point:

Patient samples- Thank you for clarifying the previous data. I believe this could be clarified further by more clearly stating that this was shown in a previous study. For example, in the following sentence by directly citing the study at this point "Importantly, this signature was not a general marker of more aggressive leukaemia, as it was (previously shown to be) not associated with resistance to conventional combination chemotherapy regimens (reference)".

Reviewer #3 (Remarks to the Author):

The authors have addressed the previous comments/concerns that I had with the manuscript and I have no further points to add.

The study represents a significant amount of work and although some of the findings are opposite to those previously obtained there is discussion regarding the potential reasons for this and it likely reflects the complexity of different model systems. It is important to publish this study for these reasons.

Anneliese Speak

Response to Reviewers

Reviewer #1 (Remarks to the Author):

The authors additions and extended discussion are appreciated. My remaining concern would be the GL261 model.

In my opinion the inability to confirm these findings robustly in a second model is of concern. The authors reference new data regarding HLA-E in a pancreatic model. Whilst this aids further support to the concept of targeting this protein to enhance CAR T therapy, it of course does not support the concept that IFN γ is overall immunosuppressive because there are obviously many of IFN γ -responsive genes that modulate the response.

However, the authors discussion place these results into the overall context of the field and I do not believe this aspect alone should preclude publication of the results.

We thank the reviewer for supporting publication of this work. We believe that the GL261 work supports the idea that IFN γ promotes CAR-T resistance, but we strongly endeavor not overstate this result. Specifically, we make it clear that the impact of interferon gamma signaling is going to vary with tumor type and microenvironment (see highlighted paragraph in the discussion). Indeed, this is a central conclusion of this study.

Minor point:

Patient samples- Thank you for clarifying the previous data. I believe this could be clarified further by more clearly stating that this was shown in a previous study. For example, in the following sentence by directly citing the study at this point "Importantly, this signature was not a general marker of more aggressive leukaemia, as it was (previously shown to be) not associated with resistance to conventional combination chemotherapy regimens (reference)".

We now provide this comment reference at the suggested location in the manuscript (highlighted).

Reviewer #3 (Remarks to the Author):

The authors have addressed the previous comments/concerns that I had with the manuscript and I have no further points to add.

The study represents a significant amount of work and although some of the findings are opposite to those previously obtained there is discussion regarding the potential reasons for this and it likely reflects the complexity of different model systems. It is important to publish this study for these reasons.

We thank the reviewer for their supportive comments